



# Barium stable isotopes as a fingerprint of biological cycling in the Amazon River Basin

Quentin Charbonnier[1,2], Julien Bouchez[1], Jérôme Gaillardet[1,3], and Éric Gayer[1]

[1]Institut de Physique du Globe de Paris, Université de Paris, CNRS, F-75005 Paris, France
[2]Institute of Geochemistry and Petrology, Department of Earth Sciences, ETH Zürich, Clausiusstrasse 25, 8092 Zürich, Switzerland
[3]Institut Universitaire de France, Paris, France

**Correspondence:** Quentin Charbonnier (quentin.charbonnier@erdw.ethz.ch)

**Abstract.** Biological cycling of rock-derived nutrients is a major component of element cycles the Earth's surface, but its magnitude to this day still remains elusive. The use of stable isotope compositions of rock-derived nutrients, which can be fractionated during biological uptake, provides a promising path forward in quantifying biological cycling and its overall contribution to global element cycling. In this paper, we investigate biological cycling in the Amazon basin using measured

elemental and stable isotopic compositions of a rock-derived nutrient, the trace element barium (Ba), in both dissolved and sediment load river samples. From these measurements, we show that dissolved Ba derives mainly from silicate rocks while a correlation between dissolved Ba and K abundances suggests a strong role of biological cycling on the Ba river budget. Further, the isotope composition of Ba ($\delta^{138}Ba$) in the dissolved load was found to be significantly different from that of the parent silicate rocks implying that dissolved Ba isotopic signatures are affected by i) formation of secondary phases and ii) biological

uptake and release from dead organic matter.

   Results from an isotope mass balance model applied to the river dissolved load data indicate that after its release to solution by rock weathering, Ba is partitioned between the dissolved load, secondary weathering products such as those found in soils and river sediments, and the biota. In most sub-catchments of the Amazon, river Ba abundances and isotope compositions are significantly affected by biological cycling. Relationships between estimates of Ba cycling and independent metrics of

ecosystem dynamics (such as Gross Primary Production and Terrestrial Ecosystem Respiration) allow us to discuss the role of erosion rates on the cycling of rock-derived nutrients.

   In addition, catchment-scale mass and isotope budgets of Ba show that the measured riverine export of Ba is lower than the estimated delivery of Ba to the Earth surface through rock alteration. This indicates the existence of a missing Ba component, that we attribute to the formation of a Ba-bearing particulate organics, possibly accumulating as soil organic matter or currently

growing biomass within the catchments; and to organic-bound Ba exported as "unsampled" river particulate organic matter.

   Given our findings on the minor nutrient Ba, we explore whether the river fluxes of most major rock-derived nutrients (K, Mg, Ca) might also be significantly affected by biological uptake or release. A first-order correction of river-derived silicate weathering fluxes from biological cycling shows that, at the Amazon at mouth, the $CO_2$ consumption by silicate weathering should be 20% higher than the yet-reported value.





Overall, our study clearly shows that the chemical and isotope composition of the Amazon (and most likely of other large rivers) bears a biological imprint, thereby challenging common assumption made in weathering studies.

## 1   Introduction

A fundamental process on Earth is the chemical weathering of rocks by atmospheric agents such as oxygen, $CO_2$, and water. Over the long-term evolution of our planet, chemical weathering reactions and the subsequent formation of secondary mineral

phases in the ocean and on the continents have been the principal drivers in forming and sustaining habitable conditions on the planets surface (Langmuir and Broecker, 2012). Chemical weathering consists of a series of hydrogeochemical processes that transform rocks within the lithosphere into dissolved solutes making up our water quality, into secondary minerals such as clays that form the diverse soils present today, and, most importantly, into the living organisms that compose our biota. (*e.g.* Porder, 2019). The complex reactions participating in these transformations occur in what is frequently referred to as

the "critical zone" of the planet, a thin layer, almost invisible at the planetary scale, made of soils, surface and underground waters as well as dead and living organic matter (Riebe et al., 2017). These weathering products at the Earth's surface are then mobilised by the water cycle and harvested by rivers before reaching the ocean. Rivers therefore integrate over the diversity of critical zone processes and offer a quantitative window into the fluxes of matter implied in these processes and their controlling factors (Berner and Berner, 2012).

An important question related to the functioning of the critical zone is the role played by living organisms in its continuous transformation and long-term evolution. Biological activity has been suggested to significantly impact the Earth surface via a wide range of processes (Brantley et al., 2011) that often tend to have a variety of different impacts on the critical zone system. For example, the decomposition of organic matter and root respiration in soils generates atmospheric $CO_2$ and organic acids that acidify the soil solution and, thus, trigger chemical weathering reactions (i.e the dissolution of primary minerals) as

a result (*e.g.* Drever and Stillings, 1997; Calmels et al., 2014). On the other hand, (*e.g.* Marston, 2010), vegetation and roots can also act as stabilizing agents by preventing physical erosion and thus preserving an optimum layer of soil. Further, one of the most prominent effects following the presence of biota on Earth is their ability to extract nutrients from rock minerals for their growth and the development of ecosystems (Napieralski et al., 2019). However, a major challenge ecosystems face is in properly recycling these extracted nutrients in a manner that is sustainable in the long-term. As a consequence, nutrient - and

more generally element - cycling by biota act as a "traffic officer" for elements and their isotopes through the critical zone, directing their circulation across the Earth surface (*e.g.* Chaudhuri et al., 2007; Cenki-Tok et al., 2009; Schmitt et al., 2012; Baronas et al., 2018). For simplicity, it has been commonly assumed that element cycling by the biota is at "equilibrium" (*e.g.* Viers et al., 2014) or, in other words, that the flux of nutrient uptake is compensated for by an equivalent flux of release (*e.g.* Gaillardet et al., 1999a). However, more and more studies are now showing that higher order plants at the ecosystem scale are

a significant reservoir of major rock-derived nutrients such as Ca, Mg, K or Si (Burghelea et al., 2018) and that nutrient cycling is a key critical zone mechanism in understanding the partitioning of elements at the Earth surface (*e.g.* Uhlig et al., 2017).





The abundance and isotope signature of rock-derived nutrients can be used to quantify biological cycling and the subsequent flux of elements passing through the critical zone. Barium (Ba), an alkali earth element, is one of these rock-derived elements, considered a minor nutrient (Viers et al., 2005), that has six stable isotopes ($^{132}$Ba, $^{134}$Ba, $^{135}$Ba, $^{136}$Ba, $^{137}$Ba, $^{138}$Ba). Besides

the formation of secondary phases such as clays or oxides (Gong et al., 2019) and adsorption (Gou et al., 2019) (considered major fractionation processes for a variety of weathering stable isotope tracers), Ba stable isotopes are also sensitive to nutrient cycling, which has been shown to drive the isotopic composition of Ba in soil water (Bullen and Chadwick, 2016). This, together with its lack of sensitivity to redox conditions (unlike transition metals or major constituents of the biosphere such as C, N, or S), makes Ba stable isotopes a potentially powerful tool to quantify biological cycling within the critical zone.

However, the study of Ba stable isotope fractionation has started only recently (Von Allmen et al., 2010) and most contributions have hitherto focused principally on seawater (Horner et al., 2015; Cao et al., 2016; Hsieh and Henderson, 2017; Bates et al., 2017; Charbonnier et al., 2018), marine sediments (Bridgestock et al., 2018) and sedimentary barite (Crockford et al., 2019). In addition, the isotope composition of Ba dissolved in continental waters has been reported only for a few rivers, such that constraints are still lacking on the global dissolved Ba riverine flux to the ocean as well as on its isotope composition (Cao

et al., 2016; Hsieh and Henderson, 2017; Gou et al., 2019).

In this study, we aim to quantify rock-derived nutrient cycling, to reveal its controlling parameters, and to discuss the potential implications thereof through the use of the abundance and isotope signature of Ba in the Amazon, the world largest river basin. We rely on the abundance and isotopic composition of river dissolved and sediment Ba for a series of sub-catchments in the Amazon Basin spanning a variety of different parent rock lithologies, reliefs, climates, and vegetation types. We test different

hypotheses to identify the principal drivers of the dissolved Ba isotopic composition of these rivers and demonstrate the significant role of biological cycling on the river Ba isotope signature. Using a set of river-scale elemental and isotopic mass budgets for Ba in the Amazon River, we further show that biological cycling has a significant impact on the routing of Ba between the different compartments of the critical zone. We then examine how this routing is dependent on the geomorphic conditions and in particular how it shifts between the highly-erosive mountainous regions of the Andes and the flatter, more

expansive areas of the Amazon Basin. Finally, we explore how the export fluxes of major dissolved species from the Amazon Basin might be impacted by biological cycling and how this might impact estimates of chemical weathering based on river chemistry. Altogether, our study opens up new perspectives for the use of Ba isotope in sedimentary archives to trace past changes in ecosystem dynamics, provides further constraints on the isotope composition of Ba delivered globally to the oceans, emphasizes the role of biological cycling on the routing of rock-derived nutrients at the Earth surface, and highlights the role

played by life on the export of materials by rivers.

## 2 Material and methods

### 2.1 Geographical and geological setting

The Amazon Basin, the world largest river basin with a drainage of 5,500,000 km$^2$ and a discharge of 6,300 km$^3$/yr (Milliman and Farnsworth, 2013), can be divided into three regions (Fig. 1):



• **The Andean belt** results from the subduction of the Nazca plate below the South American plate. The Northern part of the Amazon Andes is composed of a mixture of igneous rocks such as andesites and sedimentary rocks (*e.g.* shales and carbonates) of Paleozoic, Tertiary and Quaternary ages (Stallard and Edmond, 1983; Putzer, 1984; Moquet et al., 2011) and is drained by the Solimões tributaries, such as the Marañon, Ucayali, Huallaga, Morona, Tigre, or Pastaza. The Southern part of the Amazon Andes is composed almost exclusively of sedimentary rocks deposited during the

same geological periods as in the Northern part, and drained by the Madeira tributaries, such as the Beni, Alto Beni, Chepete, Quiquibey, Madre de Dios, Mamoré or Orthon rivers. Most Andean rivers are characterized by high erosion rates due to steep slopes, making the Andes the quasi-sole source of sediment to the Amazon mainstream (Gibbs, 1972; Allègre et al., 1996). These sediment-laden rivers, called "white waters" (Gibbs, 1967; Stallard and Edmond, 1983) also feature high dissolved loads. Rivers draining the Andean foreland (*e.g.* the Purus river), where Andean sediments have

accumulated, are also considered as "white waters", although they have lower sediment and dissolved loads compared to Andean rivers.

• **The Shields** underlie the sedimentary layers of the Amazon plains, and consist in a Precambrian craton composed of plutonic and metamorphic rocks, which crops out North (Guiana Shield) and South (Brazilian Shield) of the Amazon main channel. Amazon tributaries draining the Shields, such as the Tapajós and the Trombetas rivers, are dilute and feature

low erosion rates (Stallard and Edmond, 1983; Gaillardet et al., 1997), and are typically of the "clear water"-type (Gibbs, 1972; Stallard and Edmond, 1983), characterized by high phytoplanktonic photosynthetic activity.

• **The Amazon Plain** where all tributaries of the Solimões and the Madeira join, and underlain by sediments derived from the Andean belt and the Shields. Rivers draining only the Amazon Plain, such as the upper Negro River, are dilute, sediment-poor, and typically of the "black water"-type, *i.e.* characterized by a high dissolved organic carbon concentration. In

the Amazon Plain, Andean-fed rivers carrying sediments have developed floodplains which store river sediments over periods of time up to several $10^6$ yrs (Wittmann et al., 2011), before their release to the Amazon hydrographic system.

Throughout the remainder of the text, we will distinguish our set of sampling sites (Fig. 1) as (1) Northern Andean rivers (Solimões sub-basin); (2) Southern Andean rivers (Madeira sub-basin); (3) "dilute" rivers, lumping together rivers draining the Shields and the Amazon Plain only; (4) the "main tributaries", with sampling sites in the plain, but resulting from the mixing

of waters derived from several regions (Andes, Shields, Plain).

## 2.2 Sampling

The samples used in this study (Fig. 1) are from the Institut de Physique du Globe de Paris (IPGP) repository and have been collected during different sampling cruises performed from 2001 to 2008 (Dosseto et al., 2006; Bouchez et al., 2011; Dellinger et al., 2015b). Most analyses were performed at the High-Resolution Analytical Platform PARI of IPGP.

River waters were collected in acid-washed polypropylene containers and were filtered on site using Teflon filtration units (0.2-$\mu$m porosity). Water samples were acidified at pH ≈ 2 with ultra pure $HNO_3$ and stored in a cold room at 4°C after



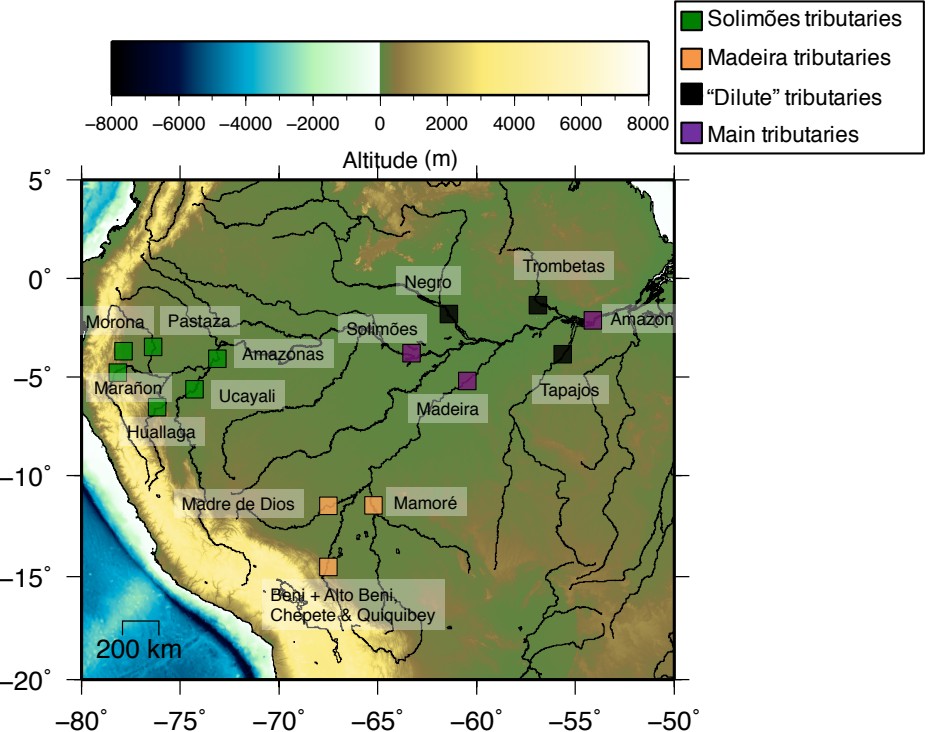

**Figure 1.** Map of the Amazon Basin and location of the sampling sites.

collection. Major anion and cation concentration were measured using ion chromatography, dissolved silica concentration by UV-VIS spectrophotometry and trace elements by quadrupole ICP-MS.

Sampling procedures for river sediments are reported in Dosseto et al. (2006) and Bouchez et al. (2011). Briefly, at several
locations, river sediments were sampled at different depths in the river channel and at the river bottom, in order to access the whole range of river sediment grain size. Major and trace element concentrations were measured by ICP-AES and ICP-MS, respectively, at the SARM (Service d'Analyse des Roches et des Minéraux, INSU facility, Vandoeuvre-les-Nancy, France; analytical details available at http://helium.crpg.cnrs-nancy.fr/SARM). In order to characterize the chemical and isotope composition of typical Andean bedrock, outcropping rocks were collected in 2001 and 2005. These rocks were crushed to obtain
homogeneous powders, digested using a mix of $HNO_3$/HF and repeatedly taken up in HCl 6 N until full dissolution. Concentration of major and trace elements in rocks were measured using quadrupole ICP-MS at IPGP.

In order to investigate the relative solubility of Ba (*i.e.* compared to soluble alkali and alkali-earth elements), we made a compilation of existing data on the abundance of Ba for rivers draining single rock types significantly present in the Amazon Basin (andesites, shales, plutonic igneous and carbonates), as well as a data compilation for the rock types themselves. The
river data is taken from a combination of literature data (Edmond et al., 1995; Louvat and Allègre, 1997; Louvat et al., 2008)



and our own new data, while rock data are derived from the GEOROC database (http://georoc.mpch-mainz.gwdg.de/georoc/) see Fig. S1 (see data Tables in repository doi:10.5281/zenodo.3698299).

## 2.3 Barium chemical separation and isotope measurements

For each river water sample, a volume of 5 to 15 mL of water was evaporated at 100°C in a Teflon beaker, and the residue was

taken up in 0.5 mL HCl 2.5 N. For each solid sample (river sediment or rock), aliquots of homogeneous powders (after crushing) were digested using a mix of $HNO_3$/HF, evaporated to dryness, and repeatedly taken up in HCl 6 N until full dissolution of the residue. A sample aliquot containing at minimum 250 ng of Ba was evaporated, and the residue was taken up in 0.5 mL HCl 2.5 N.

The chemical separation of Ba was carried out by ion chromatography using the ion-exchange AG50W-X8 resin where the

matrix was eluted in HCl 2.5 N, and Ba was eluted in HCl 6 N (Van Zuilen et al., 2016b). To ensure a proper purification of Ba from the matrix, the separation procedure was carried out sequentially twice. Ba purification from river waters and sediments does not require the use of carbonate co-precipitation, a method usually performed for Ba purification in sea water, and which can entail a loss of Ba (*e.g.* Horner et al., 2015). Therefore the double-spike method (*e.g.* Van Zuilen et al., 2016b) was not used here, and we rather corrected the mass instrumental fractionation using the sample-standard-bracketing method (see below).

However, to ensure correct and accurate Ba isotope measurements, yield and eluate purity were checked for all samples by quadrupole ICP-MS (Table S.1). The total blank procedure was checked for each separation session, with a maximum of 0.2 ng of Ba, negligible compared to a minimum of 250-300 ng (up to 300 ng) for samples.

The isotope ratios of Ba were measured using MC-ICP-MS (Thermo Fisher Neptune) coupled to a SIS spray chamber. The Faraday cups were placed to collect masses 129 (L4), 130 (L3), 131 (L2), 132 (L3), 134 (C), 135 (H1), 136 (H2), 137 (H3) and

138 (H4). The concentration of samples and standards were adjusted to 100 $\mu$g/L. Over a session, the reproducibility of the isotope measurement of a pure Ba solution was checked to be in the same range as reported by Nan et al. (2015). The isobaric interference by Xe on mass 134 was corrected by on-peak zeroes. Although the presence of residual Ce and La in the eluate can in principle lead to interferences on $^{138}$Ba, the very low concentration of these two elements in the dissolved samples did not significantly impact our measurement, as mass-dependent fractionation was shown for all dissolved load samples by

plotting $^{137}$Ba/$^{134}$Ba *vs.* $^{138}$Ba/$^{134}$Ba (Fig. S2). For solid samples, the presence of residual yet measurable amounts of Ce and La in the eluate were shown to affect the $^{138}$Ba signal to some extent. Therefore for the whole sample set, mass-dependent fractionation was rather checked using the $^{137}$Ba/$^{135}$Ba and $^{137}$Ba/$^{134}$Ba ratios (Fig. S2). For the sake of consistency, all data were measured as $\delta^{137}$Ba and are reported here as $\delta^{138}$Ba, using the mass-dependent relation $\delta^{138}$Ba $\approx 1.33 \times \delta^{137}$Ba, with the $\delta^{13x}$Ba defined as:

$$\delta^{13x}Ba_{smp} = \left( \frac{^{13x}Ba/^{134}Ba_{smp}}{^{13x}Ba/^{134}Ba_{std}} - 1 \right) \times 1000 \qquad (1)$$

with x = 7 or 8, the subscript *smp* indicating the sample isotope ratio, and the subscript *std* indicating the reference isotope ratio (NIST SRM 3104a).





Uncertainties on sample $\delta^{138}$Ba values are reported as 95%-confidence interval (CI 95%), calculated as:

$$CI95\% = t_{n-1} \times \frac{S.D.}{\sqrt{n}} \tag{2}$$

where $S.D$ is the standard deviation over the $n$ measurements of the sample (from 0.02 to 0.15‰), and $t_{n-1}$ is the student's law factor with $n-1$ degrees of freedom at a 95% confidence level. The long-term reproducibility (over one year) and accuracy of the measurements were checked using the reference materials SRM3104a (0.00±0.03‰ S.D.; $n$ = 9 from 3 chemical separations); SRM3104a spiked with two different river water matrices (-0.03±0.06‰ S.D.; $n$ = 6 with 1 chemical separation for each matrix); JB-2 (0.05±0.08‰ S.D.; $n$ = 9 chemical separations from two digestion batches) and BaBe27

(-0.81±0.08‰ S.D.; $n$ = 33 separations) matching well the values reported by Van Zuilen et al. (2016b).

## 3   Results

The concentration and isotope composition of dissolved and particulate Ba in the Amazon river system are available at doi: 10.5281/zenodo.3698299.

### 3.1   Ba abundance

The concentration of river dissolved Ba in the Amazon Basin ranges from 0.049 $\mu$mol/L to 0.490 $\mu$mol/L with an average of 0.202 $\mu$mol/L. As water flow can exert a strong control on elemental concentrations through dilution, normalization to a conservative element such as Na helps comparing dissolved Ba abundances between different water samples. The most Ba-enriched river (highest Ba/Na ratio) is the Negro river. Andean tributaries from the Solimões and Madeira basins show homogeneous Ba/Na ratios, despite the fact that they drain different rock types. The main tributaries of the Amazon (Solimões

and Madeira) show significant Ba enrichment with respect to their Andean tributaries.

Our compilation of Ba abundance (expressed as Ba/X ratios) for rocks and rivers draining single rock types (Fig S.1) highlights that Ba is depleted with respect to other alkali (Li, Na) and alkali Earth (Mg, Ca, Sr) elements during chemical weathering and thus that Ba is less soluble than these elements.

The concentration of Ba in the suspended sediments of the Amazon River Basin is heterogeneous (from 222 mg/kg to 836

mg/kg). This is likely due to the variable quartz abundance throughout the water column and a subsequent dilution effect (Bouchez et al., 2011). Once normalized to thorium (Th) concentration (Th being an insoluble element with a magmatic compatibility similar to that of Ba), Ba abundance is even more variable, with Ba/Th ratios ranging from 13 to 125. Within this range, some Ba/Th values differ significantly from the reported ratio for the Upper Continental Crust (UCC) (Ba/Th $\approx$ 50; Taylor and McLennan, 1995; Rudnick and Gao, 2003), suggesting that processes can enrich or deplete Ba in river sediment

and/or that Amazon river sediment Ba is at least partially sourced from other rock types.

The fraction of river-borne Ba exported by Amazon rivers as dissolved species is calculated as:

$$w^{Ba} = \frac{[Ba]_{diss}}{[Ba]_{diss} + [Ba]_{spm} \times [spm]} \tag{3}$$



with $[Ba]_{diss}$ the concentration of dissolved Ba in rivers in $\mu$g/L, $[Ba]_{spm}$ the Ba concentration in sediment in mg/kg, and $[spm]$ the concentration of suspended particulate matter in river (obtained by long-term sediment gauging) in g/L. The

fraction of river Ba exported as dissolved Ba ranges from 0.03 to 0.85. The lowest values are found for Andean tributaries, the highest values for "dilute" tributaries, whereas the main tributaries show intermediate values. Excluding the "dilute" tributaries, it appears that Ba is mainly transported as solids in rivers. At the mouth of the Amazon, 16 % of Ba is transported in dissolved forms.

## 3.2 Ba isotope composition

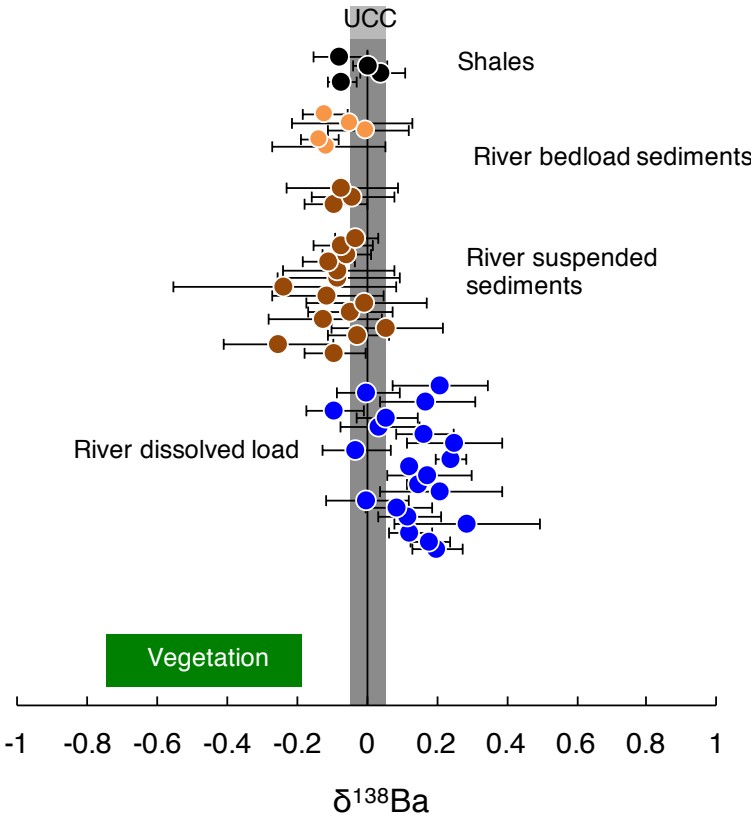

**Figure 2.** $\delta^{138}$Ba of river dissolved load, river sediments, and shale rocks from the Amazon Basin. The composition of vegetation collected over a soil profile in Hawaii (Bullen and Chadwick, 2016) and of the UCC (Upper Continental Crust; Van Zuilen et al., 2016b; Charbonnier et al., 2018; Nan et al., 2018) reported in the literature is also shown.

The isotopic composition of river dissolved Ba shows a wide range of $\delta^{138}$Ba$_{diss}$ values from 0.29‰ to -0.10‰ without any clear difference between the various types of rivers. We acknowledge that, at this stage, we lack time series of $\delta^{138}$Ba$_{diss}$





values. This could be problematic as it has been shown, for example, that $\delta^{138}Ba_{diss}$ displays temporal variations in the monsoonal Yellow River Basin (Gou et al., 2019). However, this variability was attributed to variable degrees of Ba sorption onto particles, enabled by the very high concentration of suspended particulate matter in the Yellow River (several g/L; Gou et al., 2019). As the rivers of the Amazon Basin display much lower sediment contents than that of the Yellow River (on the order of 100 mg/L for the largest tributaries; Bouchez et al., 2011), we contend that temporal variability in $\delta^{138}Ba_{diss}$ is much less significant. Nevertheless, we will return to this assumption later in the discussion.

The Ba isotope composition of river sediments, $\delta^{138}Ba_{sed}$ ranges from 0.06‰ to -0.25‰, with no significant difference between river suspended and bed sediments. The reported average values for plutonic rocks and andesites are 0.00±0.17‰ (S.D.) and 0.07±0.02‰ (S.D.), respectively (Charbonnier et al., 2018; Nan et al., 2018). Our own measurements of shale rocks collected in the Andes yield an average value for $\delta^{138}Ba_{rock}$ of -0.02±0.04‰ (S.D.) (Fig. 2). Ba in secondary weathering products is thus depleted in the "heavy" Ba isotopes with respect to the parent bedrocks (i.e. have lower isotope ratios), whereas Ba in the dissolved load is enriched in the "heavy" Ba isotopes (i.e. have higher isotope ratios). This observation is consistent with the premise that the dissolved and suspended loads are complementary phases formed during the partitioning of Ba isotopes by chemical weathering (Fig. 2 Gong et al., 2019).

The Ba isotope composition of river sediments does not show any systematic variation with respect to the Al/Si ratio (a tracer for sediment grain size), with Al-rich samples (typically surface suspended matter) enriched in fine clays and Si-rich samples (typically bedload samples) enriched in coarse quartz (Bouchez et al., 2011) (Fig. 3). This observation contrasts with what has been previously reported for other isotope systems such as Sr or Li (Bouchez et al., 2011; Dellinger et al., 2014) and highlights that Ba isotopes in solid phase weathering products are poorly sensitive to grain size.

## 4 Discussion

Significant Ba isotope fractionation is observed in the Amazon Basin between the dissolved and particulate loads of rivers (Fig. 2). In the following sections, we discuss the potential causes for this observation and show the particular influence of biological cycling on Ba isotope signatures.

### 4.1 Influence of critical zone processes on dissolved Ba

Variations in the dissolved Ba abundance and associated isotope composition could be controlled by variations in the sources of Ba and/or of critical zone processes affecting Ba after its release from rocks.

Although there are several possible sources of solutes contributing to the overall river chemistry, we show in Appendix A that silicate rocks are the main source of dissolved Ba to the river dissolved load in the Amazon Basin, consistent with previous observations from other river systems (Gou et al., 2019). This suggests that processes in the critical zone act to shift the abundance of river dissolved Ba between different basins. The fact that dissolved Ba is depleted with respect to elements classically considered as soluble (such as Na, Ca, or Mg) compared to rocks further lends support to this inference (Fig. S1).





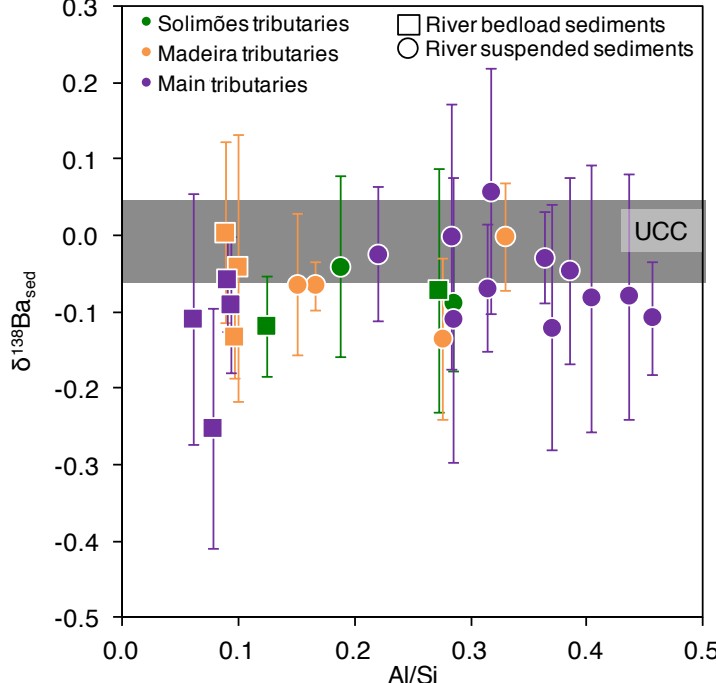

**Figure 3.** $\delta^{138}$Ba of river sediments in the Amazon Basin as a function of their Al/Si ratio, used here as a tracer for sediment grain size (Bouchez et al., 2011).

Although still largely unconstrained, it is suspected that Ba incorporation into secondary weathering products (clays or oxi-hydroxides) occurs in the critical zone (Gong et al., 2019). River dissolved Li abundance and isotope composition can serve
as a proxy for the contributions of secondary weathering products to the overall dissolved solute flux (von Strandmann et al., 2012; Dellinger et al., 2015b). When dissolved Ba is compared against dissolved Li, no correlation emerges between Ba*/Na* and Li*/Na* nor between Li*/Na* and $\delta^{138}$Ba$_{diss}$ (Fig. 4a,b) even when these elemental ratios are corrected for the source rock composition (these "corrected ratios" actually correspond to the so-called $f_{diss}^{Ba}$ and $f_{diss}^{Li}$ parameters; see below and Fig. S3). This lack of correlation implies that at least one other critical zone process (that is, in addition to the formation of secondary
weathering products) scavenges dissolved Ba after its release from rocks. This inference is also supported by the absence of any significant differences in the Ba isotope compositions between shales (which are clay- and oxide-rich sedimentary rocks), unweathered igneous rocks, and river sediments (Fig. 2), which is in contrast to what is observed for Li (Dellinger et al., 2014).

It has been suggested that the isotope composition of Ba in soils is mainly driven by biological uptake and release (Bullen and Chadwick, 2016). This can be tested using the K abundance, which is known to be strongly impacted by biological activity
at the Earth surface (Chaudhuri et al., 2007; Tripler et al., 2006; Uhlig et al., 2017; Uhlig and von Blanckenburg, 2019) - in addition to having a magmatic compatibility (Hofmann, 1988) and a high abundance in shales (Taylor and McLennan, 1995;





**Figure 4.** Scatter plots of Ba*/Na* and $\delta^{138}Ba_{diss}$ *vs.* Li*/Na* and K*/Na* for the dissolved load of the Amazon rivers ( "*" refers to dissolved concentrations corrected from non-silicate inputs, see Appendix A for details).

Rudnick and Gao, 2003) similar to those of Ba. The observation of a significant positive relationship between Ba*/Na* and K*/Na* (Fig. 4c) supports the notion of Ba behaving as a nutrient in the Amazon Basin. However, no correlation is observed between $\delta^{138}Ba_{diss}$ and K*/Na* (Fig. 4d).




Overall, the lack of correlation between $\delta^{138}\mathrm{Ba}_{diss}$ and a range of elemental ratios indicates that a combination of several processes is required to explain the Ba isotope signatures throughout the Amazon. In order to quantify the relative role of each of these processes on the river dissolved Ba budget, we first need to calculate the fraction of weathering-derived Ba that remains in solution after Ba uptake by different processes. This can be done using Na as a reference element, *i.e.* under the assumption that dissolved Na is released to rivers only by weathering and is not significantly re-incorporated in solid secondary

weathering products during their formation nor taken up as a nutrient by the biota (Gislason et al., 1996; Georg et al., 2007; Millot et al., 2010; Dellinger et al., 2015b):

$$f_{diss}^{Ba} = \frac{(Ba/Na)^*}{(Ba/Na)_0} \qquad (4)$$

with $(Ba/Na)^*$ the dissolved Ba/Na ratio corrected from non-silicate inputs, and $(Ba/Na)_0$ the bedrocks ratios (details about calculation are given in Appendix B). If we assume that Ba and Na are not fractionated during release from rocks, $f_{diss}^{Ba}$

is then a proxy for the re-incorporation of Ba into secondary weathering solids (clays, oxides) or into organic matter.

The calculated fraction of initially dissolved Ba left in the dissolved load of rivers, $f_{diss}^{Ba}$, ranges from 0.04 in rivers draining only the Andean belt, to 1.63 for the Negro River. This latter value for a parameter which in principle should be lower than < 1, is somewhat surprising, and will be discussed later (section 4.5).

$f_{diss}^{Ba}$ does not show any correlation with dissolved $\delta^{138}\mathrm{Ba}_{diss}$ (not shown), supporting the idea that several compartments

incorporate Ba in the critical zone following its release from rocks, and the various processes responsible for this incorporation are characterized by different isotopic fractionation factors. In the following, in order to quantitatively tease these effects apart, we build a model that features the two processes likely to fractionate Ba in the critical zone: secondary phase formation and biological uptake.

### 4.2   Assessing the relative role of biological uptake on the Ba cycle in the Amazon Basin

Element fluxes and isotope composition can be used in tandem to estimate the role of various processes on elemental budgets (*e.g.* Bouchez et al., 2013; Baronas et al., 2018). To this aim, we model the behavior of Ba in the critical zone at the catchment scale under a scenario in which dissolved Ba, originally derived from the partial dissolution of rocks, is partitioned into different net uptake and dissolved export fluxes following the framework developed by Bouchez et al. (2013). The partitioning of Ba after its release from rock minerals can be written as follows:

$$F_0^{Ba} = F_{diss}^{Ba} + F_{bio}^{Ba} + F_{sec}^{Ba} \qquad (5)$$

with $F_0^{Ba}$ the dissolution flux of Ba from rocks; $F_{diss}^{Ba}$ the flux of river dissolved Ba exported from the system; $F_{bio}^{Ba}$ the net Ba uptake by the biota ("net" meaning Ba uptake minus Ba release during organic matter remineralization); $F_{sec}^{Ba}$ the Ba flux associated with the net formation (*i.e.* precipitation minus re-dissolution) associated to Ba-bearing secondary weathering products such as clays or oxides.





Then dividing both sides of the eq. (5) by $F_0^{Ba}$, we obtain the relative proportion of all fluxes with respect to the dissolution flux:

$$1 = f_{diss}^{Ba} + f_{bio}^{Ba} + f_{sec}^{Ba} \qquad (6)$$

where the terms $f_i^{Ba}$ reflect the relative role of each process on Ba fluxes within the critical zone, with respect to the "initial" release flux of Ba from rocks $F_0^{Ba}$, and where $f_{diss}^{Ba}$ represents in particular the "residual" dissolved Ba flux after each process,

*i.e.* is equivalent to that in eq. (4). In other words, these terms represent the net formation fluxes of different critical zone compartments: $f_{bio}^{Ba}$ is the net uptake of Ba by the biosphere and $f_{sec}^{Ba}$ is the net flux of precipitation of Ba-bearing secondary phases of weathering. It should be emphasized that while these terms reflect the net formation of these pools, in reality the dynamics of these compartments result from the competition between their formation and their destruction. For example, clays formed deep within weathering profiles that are generally in equilibrium with deep, solute-rich waters, can be destabilized

when exposed to more dilute waters (Dellinger et al., 2015b); and organic matter can be remineralized in reservoirs with long residence times such as soils or floodplains (Bouchez et al., 2014b). A negative net formation ($f_i^{Ba} < 0$) thus indicates that the pool *i* releases more Ba to the dissolved compartment in the critical zone than it incorporates Ba from this dissolved compartment, at least on the time scale over which $f_i^{Ba}$ is estimated.

    Following Bouchez et al. (2013), we can introduce the corresponding isotope mass balance, which importantly is valid only

at steady state:

$$\delta_{diss}^{Ba} = \delta_{rock}^{Ba} - f_{bio}^{Ba} \times \Delta_{bio-diss}^{Ba} - f_{sec}^{Ba} \times \Delta_{sec-diss}^{Ba} \qquad (7)$$

with $\delta_{diss}^{Ba}$ the Ba isotope composition of the river, $\delta_{rock}^{Ba}$ the composition of the dissolved load after rock dissolution (assuming congruent dissolution, this parameter equals the isotopic composition of the average bedrock); $\Delta_{bio-diss}^{Ba}$ the Ba isotope fractionation factor during biological uptake; and $\Delta_{sec-diss}^{Ba}$ the Ba isotope fractionation factor associated with the formation

of secondary weathering products.

    In order to quantify the role of nutrient cycling in the biogeochemical dynamics of Ba at the catchment scale, eq. (7) can be solved for the term $f_{bio}^{Ba}$. Replacing $f_{sec}^{Ba}$ by (1 - $f_{bio}^{Ba}$ - $f_{diss}^{Ba}$) eq. (6) in eq. (7) and re-arranging leads to:

$$f_{bio}^{Ba} = \frac{\delta_{diss}^{Ba} - \delta_{rock}^{Ba} + \Delta_{sec-diss}^{Ba} - (f_{diss}^{Ba} \times \Delta_{sec-diss}^{Ba})}{(\Delta_{sec-diss}^{Ba} - \Delta_{bio-diss}^{Ba})} \qquad (8)$$

    Provided that the values of the parameters $\delta_{rock}^{Ba}$, $\Delta_{sec-diss}^{Ba}$, and $\Delta_{bio-diss}^{Ba}$ can be estimated, $f_{bio}^{Ba}$ can be calculated using

our measurements of $\delta_{diss}^{Ba}$ and $f_{diss}^{Ba}$ into eq. (8). In Appendix B, we discuss how the values of $\delta_{rock}^{Ba}$, $\Delta_{sec-diss}^{Ba}$, and $\Delta_{bio-diss}^{Ba}$ can be estimated. We also acknowledge that eq. (8) relies on a steady-state assumption, which is also discussed in Appendix B.

    Using these constraints, the calculated $f_{bio}^{Ba}$ values over the Amazon Basin range from -0.40 to 0.47, and $f_{bio}^{Ba}$ of the Amazon at mouth is 0.18 (Table S.2). Only a few rivers show a $f_{bio}^{Ba}$ close to 0, which challenges the following common assumption that





biological cycling operates at steady-state from the point of view of the river dissolved load (*i.e.* that nutrient uptake equals

nutrient release) at the catchment scale. Some of the rivers such as the Mamoré show a net release from the biosphere ($f_{bio}^{Ba} <$ 0), emphasizing that, locally in time and space, the biota can also act as a source of rock-derived nutrients to the river dissolved load.

The most significant findings of this analysis are the net uptake of Ba ($f_{bio}^{Ba} > 0$) by the biota across many regions of the Amazon Basin and the fact that the extent of this net biological uptake (the exact value of $f_{bio}^{Ba}$) displays significant spatial

variation. Therefore, in the following section we explore the geomorphological and ecological controls on the extent of Ba biological uptake.

### 4.3   Controls on Ba elemental and isotope partitioning by the weathering regime

Denudation rates exert a strong control on the so-called "weathering regime", which in turn influence the behavior of elements and of their isotopes in the critical zone (Bouchez et al., 2013; Dellinger et al., 2015b; Frings et al., 2016). The catchment-

scale denudation rate $D$ is the sum of the chemical weathering $W$ and physical erosion $E$ rates measured from river material, typically using estimates of weathering-derived solute fluxes (*e.g.* Gaillardet et al., 1999b) for weathering and sediment gauging for erosion. Alternatively, denudation rates can also be estimated using cosmogenic nuclides such as $^{10}$Be in sediments (*e.g.* Wittmann et al., 2011). The three rates $D$, $E$, and $W$ are typically expressed in units of t/km$^2$/y (or in mm/y using a rock-density based conversion). At one end of the spectrum of $D$, the "kinetically limited" weathering regime is characterized by

high values of $D$ and short particle residence time in the critical zone (West et al., 2005). Under these conditions, the extent of chemical reactions is limited by the kinetics (or how fast the reaction can proceed), leading to a low share of solute export ($W$ for weathering) to total export rates (low values of the $W/D$ ratios, used as an index for weathering intensity; Bouchez et al., 2014a). On the other end of the spectrum, the "supply-limited" regime, in which $D$ values are low and particles reside for a long time within the weathering reactor, allows for more intense weathering and is characterized by higher $W/D$ ratios (Riebe et al.,

2004). Weathering intensity, bounded by these "kinetically-limited" and "supply-limited" weathering regime endmembers, also translates into isotope signatures: for example, denudation exerts a strong control on the behavior of Li stable isotopes within the critical zone by modulating the extent of secondary mineral formation (Dellinger et al., 2015b; Frings et al., 2016).

Unlike for Li isotopes, there is no direct relationship between $\delta^{138}Ba_{diss}$ and $D$ or $W/D$ in our dataset (Fig. S4). This is because several critical zone processes act in conjunction to set the river dissolved Ba isotope signature in the Amazon

Basin, as explained in section 4.1, whereas only the formation of secondary products drive dissolved Li isotopes. However, the individual role of each critical zone process can be evaluated through the examination of the relationships between $f_i^{Ba}$ (eqs. (4) and (8)) fractions estimated above using our mass balance approach, and $D$ and $W/D$ as markers of the weathering regime.

The relative role of the dissolved export to the total Ba input into the critical zone ($f_{diss}^{Ba}$) displays a positive correlation with denudation rates (Fig. 5a) and a negative correlation with $W/D$ (Fig. 5b). This observation is in contrast to what has been

reported for Li, which shows a higher solubility (higher $f_{diss}^{Li}$) when $D$ is high or low, resulting in a so-called "boomerang" relationship (Dellinger et al., 2015b). For Ba, it is thus likely that even if, as for Li, Ba scavenging during formation of secondary weathering products is hampered at high $D$ because of a kinetic limitation, another process scavenging dissolved Ba is being





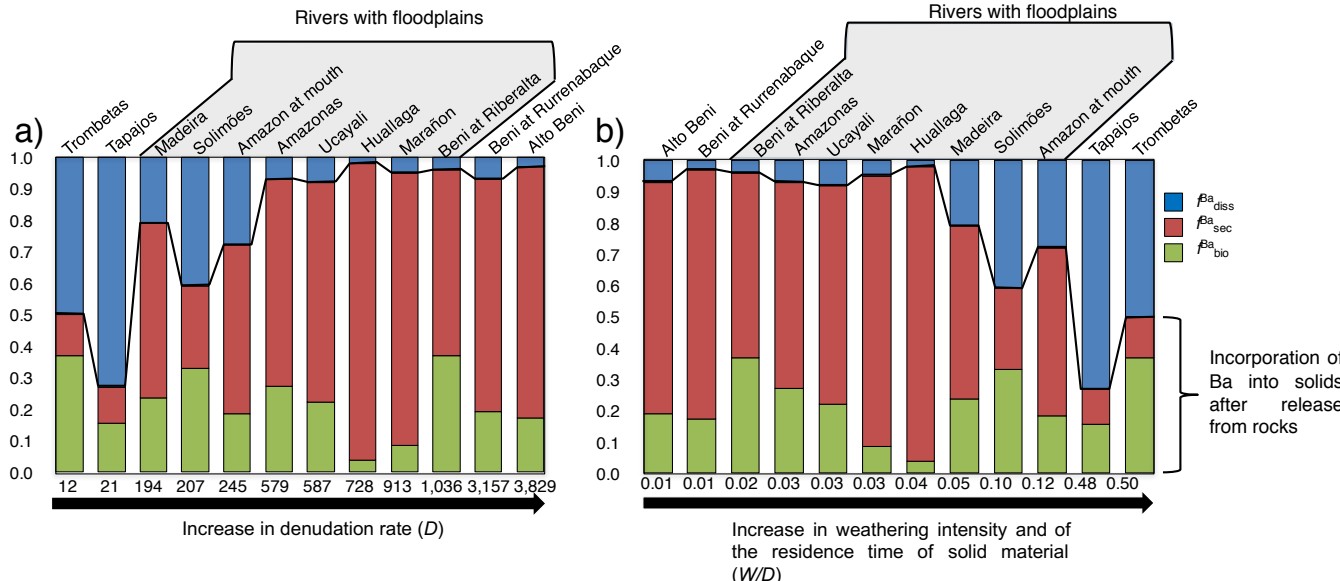

**Figure 5.** Relative role of different critical zone processes ($f_i^{Ba}$; dissolved export ($diss$), formation of secondary weathering products such as clays or oxides ($sec$), and biological uptake ($bio$); eq. (5)) on Ba fluxes within the critical zone with respect to the "initial" release flux of Ba from rocks *vs.* $D$ (denudation rate, panel a) and $W/D$ (weathering intensity, panel b), in the Amazon Basin. $D$ and $W/D$ values data are taken from Wittmann et al. (2011) and Dellinger et al. (2015b). Here, $f_{sec}^{Ba}$ is calculated as $1 - f_{diss}^{Ba} - f_{bio}^{Ba}$ (eq. (6)).

enhanced at high $D$, the nature of which is discussed below. $f_{bio}^{Ba}$ remains fairly constant and does not show any correlation with $D$ or $W/D$ (Fig. 5), whereas $f_{sec}^{Ba}$ shows a negative correlation with $W/D$ and a positive correlation with $D$ (Fig. 5).

As for each river the sum of the three $f_i^{Ba}$ values is equal to 1 by construction, some of the observed co-variation between $f_i^{Ba}$ are automatic. In order to independently examine the role of each process, we translate these Ba fractions (equivalent to non-dimensional Ba fluxes) into area-specific, dimensional fluxes $F_i^{Ba}$ (express as a kg of Ba per year per km$^2$) using the fact that by definition from eq. (6):

$$F_i^{Ba} = F_{diss}^{Ba} \times \frac{f_i^{Ba}}{f_{diss}^{Ba}} \qquad (9)$$

with $i = bio$, $sec$, or $diss$. In eq. (9) we can calculate $F_{diss}^{Ba}$ using $Q$ the river discharge (in m$^3$/yr), $[Ba]^*$ the concentration of silicate-derived river dissolved Ba ($\mu$g/L), and $S$ (in km$^2$) the drainage area of the catchment (yielding a flux in mg/km$^2$/yr that we convert in kg/km$^2$/yr; Table S.2):

$$F_{diss}^{Ba} = \frac{[Ba]^* \times Q}{S} \qquad (10)$$



The area-specific flux of dissolved Ba remains constant across the range of denudation rates (Fig. 6a) or weathering regime

indexed as the $W/D$ ratio (Fig. 6b). Barium fluxes associated with both biological uptake and the formation of secondary weathering products increase with increasing denudation rates. The flux of Ba associated to secondary phase formation is positively related to the denudation rate and, hence, to the flux of Ba linked to biological uptake. At the highest denudation rates, the formation of secondary weathering products is thought to become kinetically limited (Bouchez et al., 2013; Pogge von Strandmann and Henderson, 2015; Dellinger et al., 2015b). However, such limitation is not apparent for the corresponding Ba

flux. This indicates again that the relevant processes for scavenging dissolved elements in the critical zone are distinct between Ba and Li.

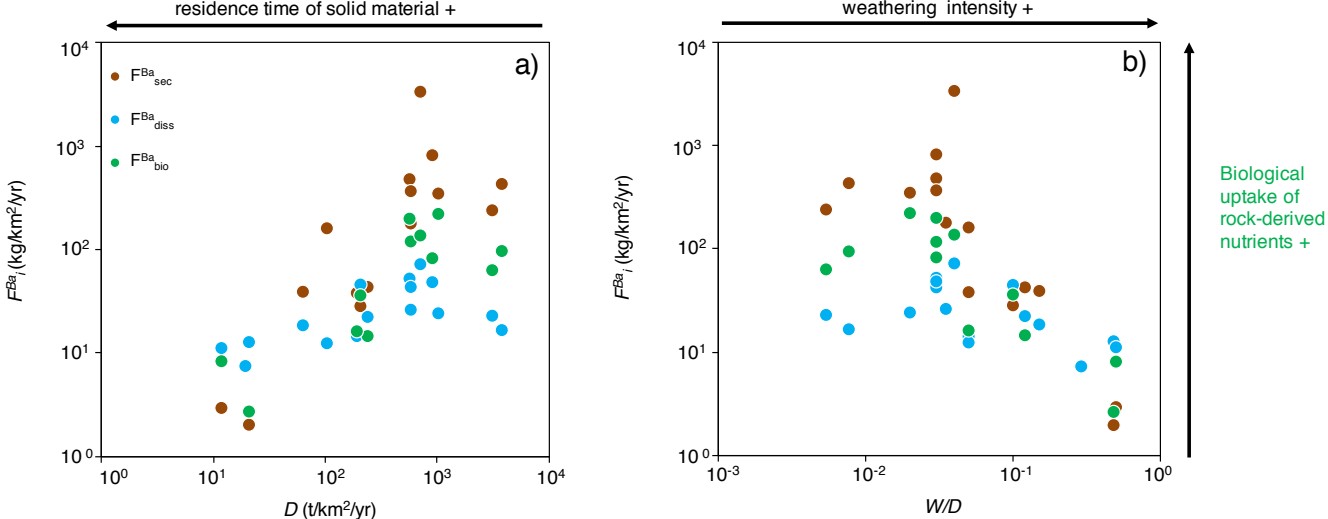

**Figure 6.** Fluxes of Ba in the critical zone ($F_i^{Ba}$: dissolved export ($diss$), formation of secondary weathering products such as clays or oxides ($sec$), and biological uptake ($bio$); eq. (9)) *vs.* $D$ (denudation, panel a), and $W/D$ (weathering intensity, panel b) in the Amazon Basin. $D$ and $W/D$ values are taken from Wittmann et al. (2011) and Dellinger et al. (2015b).

The absolute flux of dissolved export Ba ($F_{diss}^{Ba}$) being constant with $D$, combined with the fact that the fluxes of Ba associated to biological uptake and secondary phase formation increase with $D$, explains why $f_{diss}^{Ba}$ (the relative dissolved Ba flux) decreases with $D$ (Fig. 5). Regardless of the explanation for this relatively invariant $F_{diss}^{Ba}$, it is critical to note that such a

behavior, observed here for Ba across the whole Amazon Basin, is in strong contrast with what has been reported for Li in the Andean part of the Amazon Basin (Dellinger et al., 2015b). In addition, it should be emphasized that the relationships shown in Figs. 5 and 6 include pure Andean rivers (*e.g.* Alto Beni), pure "dilute" rivers (*e.g.* Tapajós), and mixed rivers with active floodplains (*e.g.* Amazon at Óbidos). Dellinger et al. (2015b) have shown a specific behavior of Li isotopes in floodplains, expressed as a strong enrichment of heavy Li isotopes in rivers featuring active floodplains, and a loss of dissolved Li between

the upstream reaches and the downstream reaches of these rivers. Although we cannot fully exclude a role of processes specific





to floodplains on the Ba cycle and isotope fractionation in the Amazon Basin, we note that no such loss of dissolved Ba is observed across the few floodplain reaches where such a calculation can be made (See Supporting text for details and Table S.3 for data; Bouchez et al., 2012; Dellinger et al., 2015b). In most floodplain reaches the Ba flux actually increases downstream, which can be explained by the relative enrichment of Ba of pure "dilute" tributaries (as shown by their relatively high dissolved
Ba/Na ratios; section 4.1) that also contribute water to large rivers featuring active floodplains.

The most striking feature of Fig. 6 remains the increase of the Ba biological uptake flux with denudation rates. Although this finding relies on isotope mass balance equations that inherently bear some uncertainties - in particular related to the steady state assumption itself, or to parameter estimates (Appendix B), such result is consistent with what has been observed at the much smaller soil scale reporting that biological cycling can be influenced by rock supply rate or age (*e.g.* Vitousek et al. (2003);
Porder et al. (2007)). In the following section, we explore the reasons why such a relationship might exist at the scale of large catchments in the Amazon Basin, in particular using independent estimates of relevant terms of the carbon cycle.

### 4.4 Relationships between biological uptake, weathering regime, and ecosystem dynamics

The fact that across the Amazon Basin, net Ba uptake ($F_{bio}^{Ba}$) is positively related to denudation rate at the catchment scale (Fig. 6) suggests that a relationship exist between the weathering regime prevailing in a catchment, and the dynamics of the
ecosystems it hosts (*e.g.* Buendia et al. (2010)). To further explore this hypothesis, we calculated two catchment-scale carbon fluxes, that we think capture the first-order dynamics of ecosystems: the Gross Primary Production (GPP; in t/km$^2$/yr) and the terrestrial ecosystem respiration (TER; in t/km$^2$/yr), both retrieved from remote sensing data (Table S.4; Tramontana et al. (2016); Jung et al. (2019), see Supporting text for details). GPP can be envisaged as a measure of the biomass - most catchment-scale biomass estimates being anyway derived from GPP estimates (Tramontana et al., 2016; Jung et al., 2019). Respiration of
organic matter leads to the release of nutrients to the ecosystem, such that catchment-scale TER can be rather used as a proxy for the recycling of rock-derived nutrients (such as Ba) through litter remineralization. These constraints do not rely on any geochemical tool, and thus offer a fully independent piece of information on the magnitude of biological activity across the Amazon Basin compared to the approach used in this study until now.

Interestingly, both GPP and TER display a positive relation with the $W/D$ ratio (Fig. 7a and 7b). These relations demonstrate
that the weathering regime, is indeed linked to the catchment-scale biological activity across the Amazon Basin. In turn, ecosystem dynamics are also related to rock-derived nutrient cycling as calculated for Ba isotopes (Fig. 7c,d), lending further confidence to the validity of our determination of $F_i^{Ba}$.

The relationships of Fig. 7 can be interpreted in the following way:

  – At high $W/D$ ratio (corresponding to poorly-erosive, tectonically-quiescent areas of the Amazon Basin such as the
Shields and Amazon Plain), the elevated GPP (Fig. 7a) suggests that the size of the biota is large (in particular due to higher temperatures in these settings), while the high TER (Fig. 7b) shows that remineralization is strong, leading to efficient recycling of rock-derived nutrients by litter remineralization, resulting in a smaller net flux associated to Ba



**Figure 7.** Gross Primary Production (GPP) (in t/km$^2$/yr) a) and Terrestrial Ecosystem Respiration (TER) (in t/km$^2$/yr) b) retrieved from remote sensing data (Tramontana et al., 2016; Jung et al., 2019) *vs.* weathering intensity ($W/D$ ratio) and c) GPP and d) TER *vs.* flux of Ba biological uptake ($F_{bio}^{Ba}$). An interpretation of these trends in terms of the relative prominence of the "geogenic" and "organic" pathways *sensu* Uhlig and von Blanckenburg (2019) is provided in panel d).

biological uptake ($F_{bio}^{Ba}$; Fig. 7d). The sustained release of bio-available rock-derived nutrients, among which Ba, would then correspond to activation of the so-called "organic pathway" of Uhlig and von Blanckenburg (2019).





– At low $W/D$ ratios (corresponding to highly-erosive settings, such as the Andes), both GPP and TER are relatively low (Fig. 7a,b), suggesting that the size of the biota reservoir is relatively small and that the efficiency of recycling of rock-derived from litter nutrients is lower, leading to a higher net flux of Ba biological uptake by the biosphere (Fig. 7d). In this regime, the "geogenic pathway" of Uhlig and von Blanckenburg (2019) prevails.

Although Fig. 7 provides further evidence for a link between ecosystem dynamics and weathering regime, the interpretation
supplied above does not provide a proper causal relationship as to why a positive relationship exists between Ba biological uptake and the denudation rate (Fig. 6). It is indeed notably difficult to disentangle the complex feedbacks between weathering, erosion, and biological cycling, which could lead to this observation (*e.g.* Brantley et al., 2011; Uhlig and von Blanckenburg, 2019; Porder, 2019). Nevertheless, in the following we propose a set of non-mutually exclusive explanations for the observed relationship between Ba biological uptake and the weathering regime.

First, the observed relation between biological uptake and denudation rate could be explained by a control of nutrient supply to the Earth surface (through its rejuvenation by denudation) on the biological uptake: the more rock-derived nutrients are made available at the Earth surface through denudation, the higher biological uptake is (*e.g* Vitousek et al., 2003; Porder et al., 2007; Hahm et al., 2014). This would imply that Ba, or some rock-derived nutrient with a behaviour in the critical zone close to that of Ba, is limiting for the development of biota growth across the Amazon Basin.

Alternatively, a larger pool of bio-available Ba (for example present on the exchange complex in soils; Bullen and Chadwick, 2016) could enhance biological uptake in high-denudation rates mountainous regions of the Amazon. This would be consistent with the stronger incorporation of Ba by secondary phases at high $D$ (shown by the positive relationship between $F_{sec}^{Ba}$ and $D$; Fig. 6), provided that this incorporated Ba remains bio-available. Why the formation of such a bio-available pool of Ba would be enhanced in highly erosive regions, where soils are thought to be thin, remains an open question that would require further
research to be evaluated. However, we note that highly erosive settings have been shown to host nutrient-rich soils (Porder et al., 2015).

The positive influence of denudation on biological uptake could also be mediated by water flow. As suggested by Torres et al. (2015), in the Amazon the residence time of water in the critical zone might be longer along steeper slopes (characteristic of mountainous settings with high denudation rates), possibly allowing for increased water availability for the biota, thereby
favouring primary production and uptake.

Finally, we hypothesize that at high $D$, efficient export of particulate organic matter by erosion, and the associated loss of nutrients, is compensated for by a stronger biological uptake and possibly by accelerated weathering (Vadeboncoeur et al., 2014; Uhlig et al., 2017). In this case, the shift in the prevailing pathway for availability of rock derived-nutrients (from the organic pathway at low $D$, high $W/D$ ratio; to the geogenic pathway at high $D$, low $W/D$ ratio; Fig. 7 and Uhlig and von
Blanckenburg, 2019), due to the efficiency of organic matter remineralisation, would in turn be itself driven by the erosion of organic matter. The latter hypothesis can actually further be tested using a so-called "river-scale mass budget", which allows for probing the form under which rock-derived nutrients are exported. Such a test is performed in the next section.



### 4.5 The fate of Ba following its biological uptake in the Amazon Basin: clues from Ba elemental and isotope river-scale mass budgets

River-scale mass budgets, based on a hypothesized chemical complementarity between the weathered (chemical export) and eroded (physical export) material exported by rivers, can be used to test whether the chemical composition of the material exported by rivers can be explained by the composition of the rock undergoing erosion and weathering at the scale of large catchments (Stallard, 1995; Gaillardet et al., 1995, 1999a). In the scope of the present study on Ba isotopes, such river mass budget can be used to test whether all Ba released from rock dissolution is found in the total (that is, dissolved plus particulate)

riverine export, in which case the mass budget can be considered to be "equilibrated". A lack of Ba in the riverine export would then mean that Ba is either currently accumulating in the catchment, and/or that the adopted sampling techniques do not capture the entirety of the riverine Ba export. As shown below, the form under which Ba is accumulated in, or exported from the catchment, can also be discussed based on the results of such a river-scale mass budget.

In addition, it is important to note that whereas strong uncertainties affect the different parameters used to compute $f_{bio}^{Ba}$ (in

particular on the isotope fractionation factors $\Delta_{sec-diss}^{Ba}$ and $\Delta_{bio-diss}^{Ba}$), river-scale mass budgets are based only on the relative partitioning of Ba and its isotopes between the dissolved and solid phases of rivers (Bouchez et al., 2013).

### 4.5.1 Testing the equilibrium of the Ba river mass budget in the Amazon Basin

To assess whether the Ba river budget is "at equilibrium" in the Amazon, we can compare the fraction of Ba transported as a dissolved species ($w^{Ba}$; eq. (3)) against the relative loss of Ba in river sediments compared to the source rocks. To quantify this

loss, we normalise the Ba/Th ratio of river sediments to the Ba/Th ratio of the source rocks (the full derivation of the following equation is presented in Appendix C).

$$(Ba/Th)_n + w_{fluxes}^{Ba} = 1 \qquad (11)$$

The two parameters $(Ba/Th)_n$ and $w_{fluxes}^{Ba}$ represent the fraction of dissolved and solid Ba transported in rivers relative to the denudation fluxes, respectively. Eq. (11) means that when the Ba river mass budget is at equilibrium, the sum of $(Ba/Th)_n$

and $w_{fluxes}^{Ba}$ is equal to 1; in other words data points should lie on the down-sloping diagonal in a $w_{fluxes}^{Ba}$ vs. $(Ba/Th)_n$ diagram (Fig. S5 and 8a). Rivers plotting below this line are characterized by an export of Ba lower than that expected from the basin-scale denudation rate and the Ba concentration in the source rock.

Such a test for an equilibrated river mass budget can also be performed using isotope ratios (Bouchez et al., 2013; Uhlig et al., 2017):

$$w_{iso}^{Ba} = \frac{\delta_{rock}^{Ba} - \delta_{sed}^{Ba}}{\delta_{diss}^{Ba} - \delta_{sed}^{Ba}} \qquad (12)$$

In a $w_{iso}^{Ba}$ vs. $w_{fluxes}^{Ba}$ diagram, if the river mass budget is at equilibrium the data points should lie along the 1:1 line, while in a $w_{iso}^{Ba}$ vs. $(Ba/Th)_n$ diagram, they should lie along the down-sloping diagonal. Note that the $w_{iso}^{Ba}$ vs. $(Ba/Th)_n$





evaluation, unlike the $w_{fluxes}^{Ba}$-$(Ba/Th)_n$ comparison explained above, does not depend on fluxes, but only on the intensive

metrics $(Ba/Th)_n$ and $w_{iso}^{Ba}$. In order to provide an example of an equilibrated river mass budget, in Fig. S5 we re-visit the

data provided by Dellinger et al. (2015a), who showed that in the Amazon the sum of the solid and dissolved Li fluxes exported

by rivers, as well as their combined isotope composition, can be accounted for by independent estimate of flux and isotope

composition of Li eroded and weathered from rocks.

The comparison between $w_{fluxes}^{Ba}$ and $(Ba/Th)_n$ (eq. (C5); Fig. 8a) shows that in most of the Amazon rivers, there is

less Ba exported by rivers than expected from denudation rates. Notable exceptions are the "dilute" tributaries (Rio Negro,

Trombetas and Tapajós). It is possible that in such rivers, the export of Th as a dissolved species induced by organic colloids

compromises the use of eqs. (C2) and (C3). We also note that in these lowland areas, the input from dust (Abouchami et al.,

2013; Moran-Zuloaga et al., 2018) can be significant, therefore biasing our estimates for the river-scale mass budgets based

on (C1) or (C2). In the following, we focus our interpretations on the other river systems, which all display a negative offset

from the equilibrium line in Fig. 8a. For these rivers, we note that if a dust component was included in the left-hand sides of

(C1) or (C2), the observed disequilibrium would go in the same direction and would even be larger, thus leaving this inference

unchanged.

The observed disequilibrium in the Ba river mass budget also remains when comparing $w_{iso}^{Ba}$ (eq. (12)) to $(Ba/Th)_n$ (eq.

(C5), Fig. 8b) or $w_{fluxes}^{Ba}$ to $w_{iso}^{Ba}$ (Fig. 8c), despite significant uncertainty on the values of $w_{iso}^{Ba}$ due to both analytical un-

certainties and error propagation (for example when $\delta_{diss}^{Ba}$ and $\delta_{sed}^{Ba}$ are close; see eq. (12) and discussion in Bouchez et al.,

2013).

From here on, in order to identify the potential causes for the lack of equilibrium in the river-scale mass budgets observed for

most tributaries of the Amazon Basin, we can test the different scenarios that could reconcile the isotopic and elemental exports

of Ba by Amazon rivers with denudation rates. As there is less Ba exported by rivers than expected from denudation (Fig. 8a),

these different scenarios will all consider that to reconcile the two measures, addition of a "missing Ba component", associated

with a specific isotopic signature called hereafter $\delta_{miss}^{Ba}$, is needed. To illustrate these scenarios, arrows in the panels of Fig. 8

indicate in which directions the values on the three diagram axes will change upon addition of the missing Ba component:

**1)** the missing Ba is under a dissolved form and its isotopic signature ($\delta_{miss}^{Ba}$) is lower than $\delta_{diss}^{Ba}$

**2)** the missing Ba is under a dissolved form and $\delta_{miss}^{Ba}$ is higher than $\delta_{diss}^{Ba}$

**3)** the missing Ba is under a particulate form and $\delta_{miss}^{Ba}$ is lower than $\delta_{sed}^{Ba}$

**4)** the missing Ba is under a particulate form and $\delta_{miss}^{Ba}$ is higher than $\delta_{sed}^{Ba}$

For scenarios 1 and 2, adding the missing (dissolved) Ba would increase $w_{fluxes}^{Ba}$ (eq. (3)) but leave $(Ba/Th)_n$ unchanged.

Scenarios 3 and 4 would result in a decrease of $w_{fluxes}^{Ba}$, while the shift they would induce on $(Ba/Th)_n$ would depend on

the Ba/Th ratio of the missing Ba component. We contend that this ratio is relatively high, as we expect - as discussed below

in section 4.5.2 - that this missing Ba solid component would be of biological nature, and as Th is not a nutrient. In this case,

scenarios 3 and 4 would result in the addition of Ba-rich material to the river solid load, and thus to an increase in $(Ba/Th)_n$,





leading to non-vertical vectors in Fig. 8b and Fig. 8c. Finally, following eq. (12), $w_{iso}^{Ba}$ would increase if $\delta_{miss}^{Ba}$ is higher than $\delta_{diss}^{Ba}$ (scenario 1) or than $\delta_{sed}^{Ba}$ (scenario 3); and decrease if $\delta_{miss}^{Ba}$ is lower (scenarios 2 and 4).

While by construction all these scenarios bring the datapoints closer to the equilibrium line in Fig. 8a, it appears that scenario 2 can be immediately excluded because adding dissolved Ba with a high $\delta_{miss}^{Ba}$ would shift the river mass budget toward stronger 510 disequilibrium (Fig. 8b and c). In addition, the direction given by the vector of scenario 4 is also unlikely to reconcile $w_{iso}^{Ba}$ and $(Ba/Th)_n$ for any river in Fig. 8b and $w_{iso}^{Ba}$ and $w_{fluxes}^{Ba}$ for most rivers in Fig. 8c. Indeed, some rivers (Beni at Riberalta and Rurrenabaque, Ucayali and Mamoré) are above the equilibrium line in Fig. 8c, and their Ba river mass budget could be reconciled if the vector 1 had a sub-horizontal direction, which could be the case if the difference between $\delta_{diss}^{Ba}$ and $\delta_{miss}^{Ba}$ was small. However, we first note that for these rivers the uncertainty on $w_{iso}^{Ba}$ is large. Second, the most important finding regarding 515 this comparison between $w_{iso}^{Ba}$ and $w_{fluxes}^{Ba}$ in Fig. 8c is that the observed disequilibrium cannot be explained by scenario 2 and is unlikely to be explained by scenario 4.

To summarize, ruling out scenarios 2 and 4 allows us to infer that the isotopic composition of the missing Ba ($\delta_{miss}^{Ba}$) is lower than the one of Ba exported by rivers. Therefore, the remaining valid scenarios are 1 (the missing Ba is dissolved) and 3 (the missing Ba is solid). However, the nature of (and the uncertainties encountered in) the analysis performed above prevents 520 a full assessment of the exact nature of this missing Ba (dissolved or solid), and of its isotopic composition.

Nevertheless, the difference in the Li (Fig. S5) and Ba (Fig. 8) river mass budgets, and the fact that the Ba isotopic composition of the missing reservoir is low (hence consistent with the Ba isotope composition of a biological component; Bullen and Chadwick, 2016, and Fig. 2) are strong arguments towards a role of biological cycling in forming this Ba component missing from the river-scale mass budget. In the following, we further explore this hypothesis by comparing estimates of the amount of 525 missing Ba to other independent estimates of Ba fluxes across the Amazon Basin.

### 4.5.2   What is the nature of the "missing Ba" component in the Amazon riverine export?

Given the strong suspicion of a role of the biological uptake on Ba abundance and isotope ratio in the rivers of the Amazon Basin, here we test for the influence of biological uptake on the "missing Ba" inferred from the river mass budget above.

First, we quantify the missing Ba as a relative flux ($f_{miss}^{Ba}$) as the difference between the Ba flux actually exported by rivers 530 and the Ba export that should result from denudation and the concentration of Ba in rocks. This actually corresponds to the graphical offset (vertical or horizontal) between $(Ba/Th)_n$ (or equivalently $w_{fluxes}^{Ba}$) and the down-sloping diagonal line in Fig. 8:

$$f_{miss}^{Ba} = 1 - w_{fluxes}^{Ba} - (Ba/Th)_n \tag{13}$$

It is important to note that $f_{miss}^{Ba}$ can be interpreted as an actual flux of Ba exported from the catchment but missed by our 535 sampling scheme; or as a rate of Ba build-up into a reservoir within the catchment; or a combination of thereof. We return to this distinction later when discussing the interpretation of $f_{miss}^{Ba}$, but name this metric a "flux" for the moment, for the sake of



**Figure 8.** Test for the equilibrium of the river mass budget of Ba in the Amazon Basin using elemental (eqs. (3) and (C4)) and isotopic (eq. (12)) ratios. Uncertainties on $w_{fluxes}^{Ba}$ stem from the two ways this parameter can be calculated (river gauging and cosmogenic nuclides; see Appendix C). Uncertainties on $w_{iso}^{Ba}$ have been evaluated using Monte Carlo error propagation based on the uncertainties on individual parameters in eq. (12), and are shown here as 68%CI. In panel c, "B@Rur", "B@Rib", "U", "M", "A" and "S" reflect the sampling sites: Beni at Rurrenabaque, Beni @ Riberalta, Ucayali, Madeira, Amazon, and Solimões, respectively.



simplicity. To compare the flux of missing Ba ($f_{miss}^{Ba}$) with the amount of Ba cycled by the biota in the Amazon ($f_{bio}^{Ba}$), we first convert $f_{miss}^{Ba}$ into a dimensional flux:

$$F_{miss}^{Ba} = D \times f_{miss}^{Ba} \times [Ba]_{rock} \tag{14}$$

with $F_{miss}^{Ba}$ the area-specific flux of missing Ba in rivers, expressed in kg/km$^2$/yr. We note that the denudation rates derived from the sum of sediment gauging ($E$) and silicate weathering flux ($W$) *vs.* cosmogenic nuclides which already represents the total denudation flux (Wittmann et al., 2011; Dellinger et al., 2015a) can display differences (especially in the Beni River system). Therefore, for rivers where both constraints are available, we calculate two values of $F_{miss}^{Ba}$: one based on $D$ calculated as $E$ from sediment gauging, plus the silicate weathering flux; and the other based on $D$ determined from cosmogenic nuclides.

The missing flux calculated from eq. (14) $F_{miss}^{Ba}$, is positively related to $F_{bio}^{Ba}$ (Table S.6; Fig. 9) further supporting that disequilibrium in the river mass budget across the Amazon is related to biological uptake. Interestingly, this relation is stronger when $F_{miss}^{Ba}$ is calculated using cosmogenic isotopes in eq. (14). The cosmogenic-derived denudation rates appear to integrate over longer timescales compared to that of sediment gauging (von Blanckenburg, 2005), suggesting that the link between Ba biological uptake and the Ba river mass budget is a long-term feature of the Ba cycle in the Amazon Basin. Altogether, these

observations further support that the existence of the missing Ba flux in the Amazon Basin, inferred from the river-scale mass budget performed in section 4.5.1, is indeed linked to biological cycling.

A first possibility to account for such a missing Ba flux could derive from groundwater contributions that can export a significant amount of dissolved Ba. Thus, sampling schemes focusing only on river export might miss this potentially large dissolved Ba flux coming from the groundwater. However, with the magnitude of the river Ba dissolved flux and that of $F_{miss}^{Ba}$

in the Amazon Basin being similar (Table S.6) coupled with observed Ba concentration in the groundwater being similar to those measured in the rivers, this explanation would require that the groundwater discharge of the Amazon River is similar to that of the river discharge. Although seasonal groundwater storage in the Amazon Basin appears to be a significant component of the regional hydrological budget (Frappart et al., 2019), such a high contribution of groundwater to the overall water export of the whole Amazon basin seems unlikely. In addition, we do not see any particular reason why such a groundwater export

would affect Ba and not Li (Fig. 8 and Fig. S5). It also appears unlikely that $F_{miss}^{Ba}$ could be interpreted as a dissolved pool of Ba accumulating in the catchments given the typical residence times of groundwater even in large catchments and the fact that the observed disequilibrium in the Ba river mass budget seems to hold over long time scales (see above).

Therefore, the most likely interpretation for the existence of a significant missing Ba is the export (if interpreted as an actual flux) or accumulation (if interpreted as a pool building up) of Ba-bearing solids. Given the relationship observed between $F_{bio}^{Ba}$

and $F_{miss}^{Ba}$ (Fig. 9), it is plausible that these particulates are of organic nature, *i.e.* living beings, or dead organic matter (litter and/or river particulate organic carbon). In support of this interpretation, we note that there exists a relationship between TER *vs.* $F_{miss}^{Ba}$ (Fig . 10), showing a more pronounced Ba missing flux, in basins for which the efficiency of the remineralization is lower and, thus, where the formation (and possibly export) of litter and dead particulate organic matter is stronger.


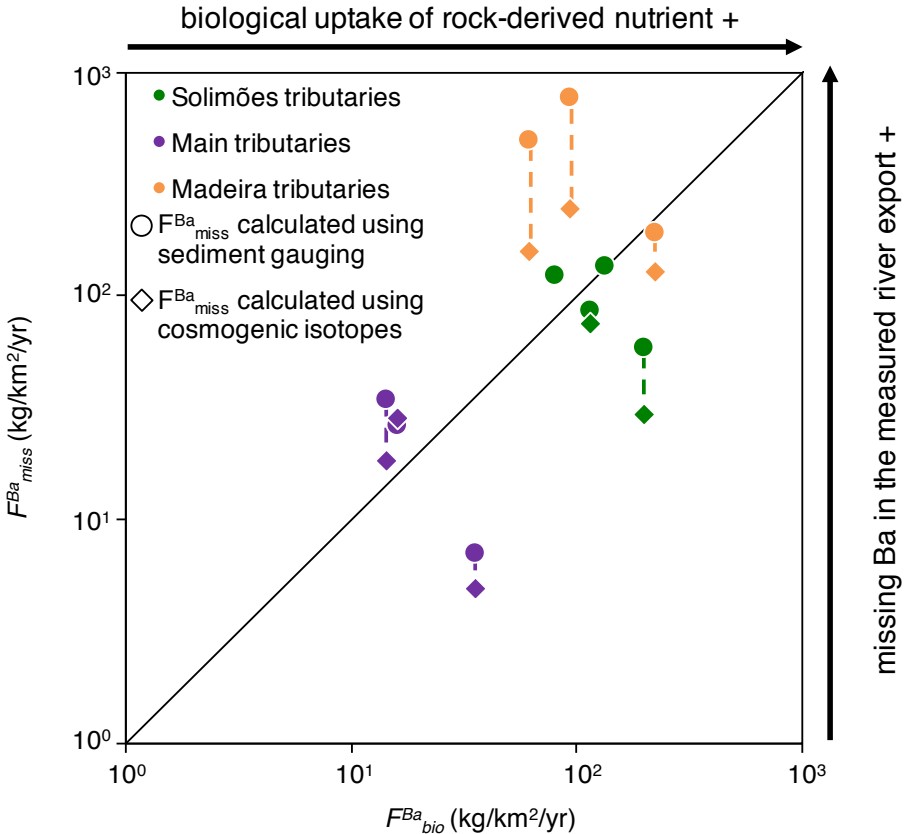

**Figure 9.** Flux of Ba biological uptake ($F_{bio}^{Ba}$) *vs.* the missing Ba flux as quantified from the river-scale mass budget ($F_{miss}^{Ba}$) in the Amazon Basin.

Accumulation of organic-bound Ba within catchments could explain the existence of the missing Ba. For example, a pool

of solid, organic-bound Ba could be forming through accumulation of litter in soils, or through increase in the biomass - this can be for example the case for area where forests regrowth after important deforestation in the Amazon Basin. Although this hypothesis needs to be tested further, it has been reported that for some regions of the Andes the biomass was currently growing, which would support this scenario (*e.g.* Feeley and Silman, 2010).

Alternatively, the missing Ba could also reflect the existence of a particulate, organic debris flux. Uhlig et al. (2017) have

suggested that the export of Mg and other rock-derived nutrients as particulate organic matter in mountain belts could represent a crucial missing riverine elemental flux, but is traditionally challenging to sample, notably because this export is mostly episodic, and, thus, difficult to constrain. In large river systems such as the Amazon, the effect of episodic export on the total organic matter flux is likely to be smeared out, as suggested by the relative regularity of particulate organic carbon fluxes throughout the hydrological cycle (Bouchez et al., 2014b). However, large woody debris (*e.g.* trunks) or river macrophytes

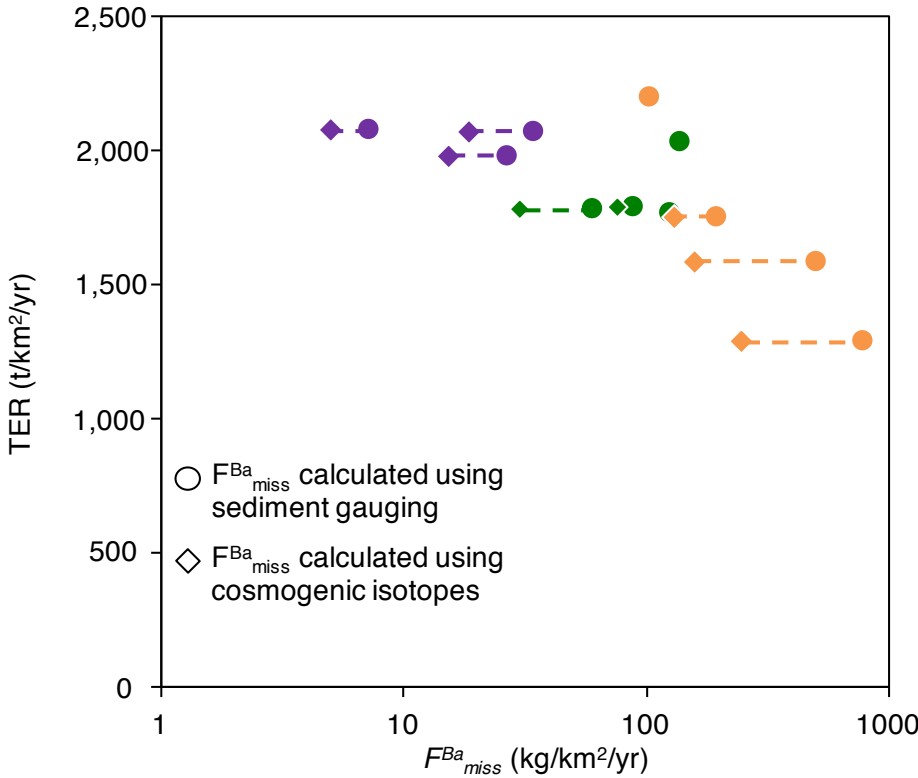

**Figure 10.** Terrestrial Ecosystem Respiration (TER) retrieved from remote sensing data (Tramontana et al., 2016; Jung et al., 2019) *vs.* the missing Ba flux ($F^{Ba}_{miss}$).

could in principle represent a significant fraction of the particulate export of large rivers and be essentially unaccounted for due to limited amount of water that can be withdrawn with the sampling scheme commonly used (only a few liters with river point samplers) (Bouchez et al., 2011).

To test for the validity of this scenario, we can predict the flux of particulate organic carbon that would be needed to account for the computed $F^{Ba}_{miss}$:

$$F^{C}_{miss} = \frac{F^{Ba}_{miss}}{[Ba]_{org}} \qquad (15)$$

with $F^{C}_{miss}$ the missing flux of organic carbon in t/km²/yr and $[Ba]_{org}$ the concentration of Ba in organic matter, estimated to be around 20 mg/kg following our own compilation (Li, 2000; Chiarenzelli et al., 2001; Bullen and Bailey, 2005; Vaganov





et al., 2013, ; Table S.7). $F_{miss}^{C}$ can then be compared with $E_{org}^{C}$, the flux of particulate organic carbon exported by the river in t/km²/yr:

$$E_{org}^{C} = E \times POC \tag{16}$$

with POC representing the relative content of organic carbon in the river sediments (Bouchez et al., 2014b), and $E$ the erosion rate (from sediment gauging) in t/km²/yr (Table S.6).

The comparison between $E_{org}^{C}$ (eq. (16)) and $F_{miss}^{C}$ (eq. (15)) suggests that almost 99% of the particulate organic carbon would be missed when estimating the POC flux through the "classical" sampling scheme. We believe that it is unlikely that such a high proportion of the POC flux can be unaccounted for. However, we note that our estimate of $[Ba]_{org}$ is based on data from living organic matter, and that it is plausible that the Ba concentration in soil / river particulate organic matter increases as organic matter undergoes partial remineralization (*e.g.* Horner et al., 2017) if, for example, Ba is bound to the more refractory fraction of the soil / sediment organic molecules. This would make our estimate of the needed missing POC flux lower. Clearly, more work is required to evaluate this possibility (Riotte et al., 2014).

Although the exact form of the missing Ba (*i.e.* flux *vs.* pool, or a combination thereof) inferred from the river-scale mass budget remains hard to ascertain, the most likely interpretation for the existence of this missing source is the partitioning of Ba into biological material after uptake. As a consequence, our analysis emphasize the role of biological cycling on the routing and transport of rock-derived nutrients at the large catchment scale.

To summarize the above discussion, this examination of the Ba elemental and isotope budgets for a range of different river catchments throughout the Amazon Basin shows how stable isotope signatures can be used to explore nutrient cycling in the critical zone and, further, opens the possibility for an application to other isotope systems and other locations. Additionally, applying this approach to Ba provides novel information on the dynamics of the rock-derived nutrients, and possible of organic matter, at the scale of large catchments.

### 4.6 A conceptual model for the influence of biological cycling and weathering regime on the fate of rock-derived nutrients in the Amazon Basin

To summarize our analysis, we propose the following interpretation for the behavior of Ba as a rock-derived nutrient in the Amazon Basin as a function of the weathering regime (Fig. 11):

– Except for some rivers draining the lowlands (see below), at low denudation rates and high weathering intensity litter remineralization outpaces litter erosion, allowing the biosphere to efficiently recycle nutrients such as Ba from the litter and to minimize the uptake of "new" Ba derived from rock dissolution (as shown by the weak net Ba uptake). In such conditions, primary production is high, promoted by higher temperatures and soil stability, and the export of particulate organic carbon from the catchment (or its accumulation within the catchment) - and associated rock-derived nutrients - is low.




– At high denudation rates and low weathering intensity, nutrient recycling by vegetation from litter is slower than litter erosion. As a consequence, the biota continuously extracts nutrients such as Ba from the surrounding rocks, which compensates for the loss of litter by erosion. Primary production is low because the growth of biota is limited by the strong denudation rates and short residence times of material, whereas export of rock-derived nutrients by particulate organic matter from the catchment (or accumulation within the catchment) is high.

– Some rivers draining the lowlands (also characterized by low denudation and high weathering intensity; "dilute" tributaries) are enriched in Ba with respect to the source rocks, which implies an export of dissolved Ba after release from the biomass / organic matter component of the critical zone and, thus, suggests that these pools are currently not at steady state.

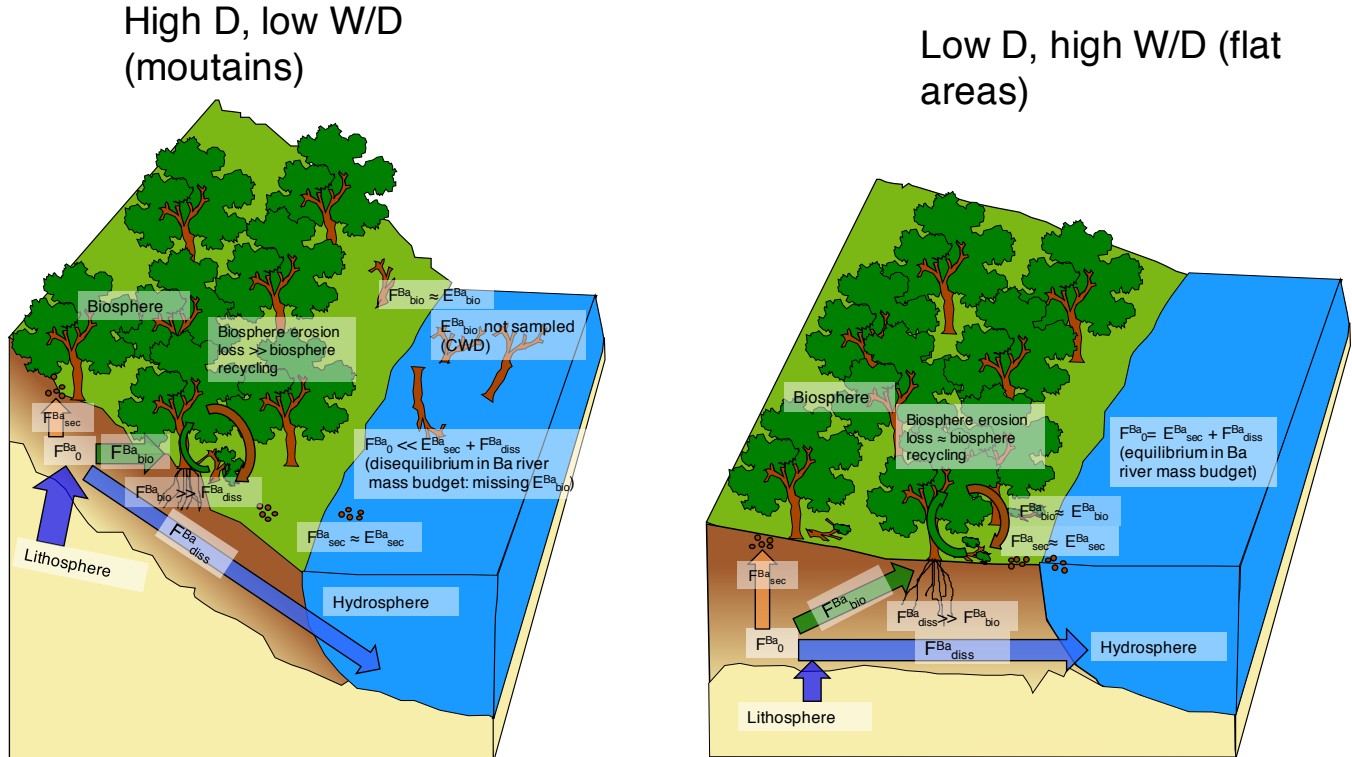

**Figure 11.** Conceptual sketch for the Ba cycle in the critical zone as a function of the weathering regime in the Amazon Basin. $E_{sec}^{Ba}$ and $E_{bio}^{Ba}$ correspond to the erosion flux of secondary phase of weathering and organic-bound Ba, respectively. Other terms are defined in the text. For the sake of simplicity, the formed particulate organic carbon pool containing rock-derived nutrients such as Ba is assumed to be exported rather than accumulated in the basin.





### 4.7 Using Ba stable isotopes to refine the river mass budget of major elements

The above analysis shows that the cycling of Ba (and those of other rock-derived nutrients) by land biota is a significant

component of terrestrial biogeochemical cycles. As a consequence, biological cycling within a catchment and the routing, or exchange, of rock-derived elements amongst the different compartments of the critical zone could bear consequence on the dissolved riverine export of major nutrients (Ca, Mg, K) that are traditionally used to estimate alkalinity fluxes and $CO_2$ consumption by chemical weathering at the catchment scale. In the following, we provide an assessment for the river-borne cations flux deriving strictly from silicate weathering and, thus, corrected from biological cycling and partitioning of major

nutrients into solid organic matter "missing" from the riverine export (see Appendix D for details).

First noting that for any nutrient $X$, the uptake flux of this element into the biota is:

$$[X]_{bio} = [Ba]_{diss} \times \frac{f_{bio}^{Ba}}{f_{diss}^{Ba}} \times \left(\frac{X}{Ba}\right)_{bio} \qquad (17)$$

with $[X]_{bio}$ the concentration of a major soluble element $X$ (Ca, K or Mg, $[Na]_{bio}$ being negligible as Na is not a nutrient) retained by the biosphere, and $(X/Ba)_{bio}$ the X/Ba ratio of the organic component. The $(X/Ba)_{bio}$ ratio is obtained from

our new compilation of the chemical composition of the vegetation (Li, 2000; Chiarenzelli et al., 2001; Bullen and Bailey, 2005; Vaganov et al., 2013, ; Table S.7). $[X]_{bio}$ can then be added to the silicate-derived dissolved river concentration of $X$ in the calculation of "corrected" silicate weathering fluxes. To that aim, we express the impact of biological cycling on silicate weathering fluxes as the ratio between cation deriving from silicate weathering added from that deriving from biological cycling ($W_{silcorr}$) and uncorrected cation ($W_{sil}$):

$$R_{(sil+bio)/sil} = \frac{W_{silcorr}}{W_{sil}} = \frac{2 \times \left([Ca]_{sil}^{2+} + [Ca]_{bio}^{2+}\right) + 2 \times \left([Mg]_{sil}^{2+} + [Mg]_{bio}^{2+}\right) + \left([K]_{sil}^{+} + [K]_{bio}^{+}\right) + [Na]_{sil}^{+}}{2 \times [Ca]_{sil}^{2+} + 2 \times [Mg]_{sil}^{2+} + [K]_{sil}^{+} + [Na]_{sil}^{+}} \qquad (18)$$

$W_{sil}$ is calculated using the sum of the concentration of each X element from silicate weathering and is expressed in meq/L, with $W_{sil} = 2 \times [Ca]_{sil}^{2+} + 2 \times [Mg]_{sil}^{2+} + [K]_{sil}^{+} + [Na]_{sil}^{+}$ (data from Dellinger et al. (2015b)). As our primary interest is to quantify the effect of biological cycling on $CO_2$ consumption, we do not consider the $SiO_2$ component of the weathering flux in our analysis. $R_{(sil+bio)/sil}$ is plotted *vs.* the $W/D$ ratio in Fig. 12. Biological cycling influences the river cation load differently

depending on the weathering regime. The highest $R_{(sil+bio)/sil}$ is observed at $W/D$ close to 1 whereas at intermediate $W/D$ (0.05) $R_{(sil+bio)/sil}$ is lower (close to 1, corresponding to a good agreement between $W_{silcorr}$ and $W_{sil}$, hence to a negligible effect of biological cycling on cation export fluxes). At low $W/D$ ratios, $R_{(sil+bio)/sil}$ exhibits a higher value (from 1.2 to 1.4), although more samples at this weathering regime would be needed to further support this observation.

First, the notably strong influence of biological cycling on silicate weathering fluxes (mainly for K) at high $W/D$ is likely

due to the fact that, under these conditions, both biological uptake and weathering rates are low such that if biological uptake is slightly higher than the slow weathering rate, the $R_{(sil+bio)/sil}$ ratios would shift to higher values. At intermediate $W/D$ values (0.05), $R_{(sil+bio)/sil}$ is the lowest suggesting that even though biological uptake is relatively strong compared to the





high-$W/D$ regime, the weathering rate is much higher and, thus, biological cycling only exerts a relatively small influence on the overall silicate weathering fluxes. Finally, $R_{(sil+bio)/sil}$ is shown to increase at very low $W/D$, implying that when

denudation rates are very high, biological uptake increases more rapidly (allowing for the replacement of the eroded biospheric component) than weathering rates. The fact that biological cycling can strongly influence the cation dissolved load at low $W/D$, corresponding to high $D$, challenges the traditional view about chemical weathering regimes in kinetically-limited conditions. Silicate weathering rates are thought to reach a maximum at a finite denudation rate (West et al., 2005; Dixon and von Blanckenburg, 2012). However, given our results, it appears possible that weathering rates at high $D$ have hitherto been

underestimated as a result of maximum biological uptake. Further, research on biological cycling at high denudation rates is necessary to shed light on the possible links between weathering rates and denudation rates when the influence of vegetation and the surrounding biota is taken into account.

    For the entirety of the Amazon Basin, our findings suggest that the silicate weathering flux (and the associated $CO_2$ consumption) is underestimated by 20%. However, we would like to stress that this result is subjected to significant uncertainty

and that to refine this first-order observation other tracers of nutrient cycling should be used to obtain a more accurate value.

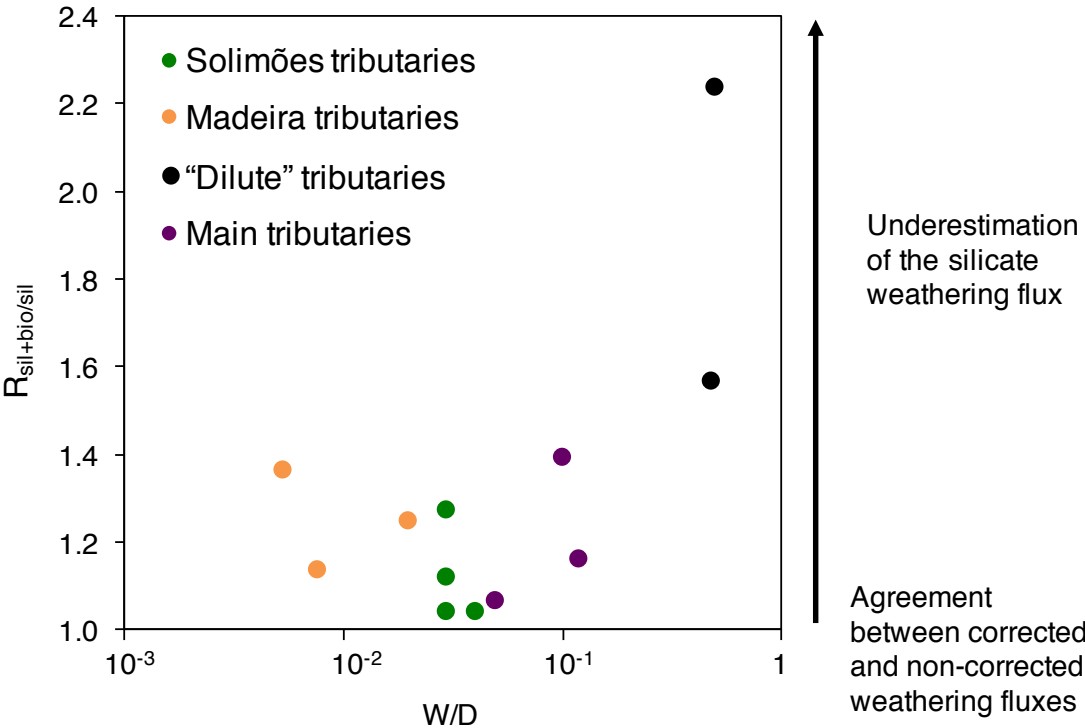

**Figure 12.** $R_{(sil+bio)/sil}$ (difference between the biological-uptake silicate weathering fluxes-corrected and the uncorrected silicate weathering flux calculated from the river dissolved load) *vs.* $W/D$ (weathering intensity).





# 5   Conclusions

In the Amazon, the abundance and isotope composition of Ba - which we show to be sourced only from silicate rocks - is controlled by a combination of the formation of secondary weathering products and biological uptake, with a stronger isotope effect of the latter.

Using isotope mass balance and estimates of isotope fractionation factors, we are able to propose a quantitative pattern for the partitioning of Ba liberated in the critical zone through rock weathering amongst secondary phases of weathering, the dissolved load, and the biota. The comparison of this partitioning to catchment-scale mass and isotope budgets of Ba (showing that the riverine export of Ba does not match the flux of Ba release from rocks) suggests that biological cycling of Ba leads to the formation of a "missing Ba component" in the critical zone of the Amazon Basin. Although the exact nature of this

missing Ba component remains elusive (in particular regarding the fact that it could be accumulated within the catchment and/or exported by rivers), our favored interpretation is that it is locked into dead particulate organic matter (*i.e.* litter or river particulate organic carbon).

The proportion of Ba remaining in solution compared to that "initially" released by mineral dissolution is negatively correlated to denudation rates and positively correlated to weathering intensity, reflecting the combined effects of scavenging by

secondary mineral precipitation and biological uptake. Biological uptake is found to be stronger at high denudation, possibly reflecting the need for the biota to compensate for the loss of organic components lost by erosion. When the denudation rate is low, organic-bound Ba can be remineralized such that additional biological uptake is not required to replace the solid organic material lost from the system.

Finally, using the estimated uptake of Ba by the biosphere, we re-assess silicate weathering fluxes calculated from the cation

loads of rivers with the effect of biological uptake now corrected for. We show that the cation dissolved load at the mouth of the Amazon is underestimated by 20%, which is enough to shift estimates of $CO_2$ consumption deduced from the cation dissolved load. This observation should be confirmed by using other rock-derived nutrients to reduce the uncertainties. Further, we believe that the method developed in this study to account for contributions from biological cycling should be extended to other large rivers in order to re-assess the $CO_2$ consumption at a global scale.

To conclude, our work shows that barium and its isotopes can be used to track biological cycling of rock-derived nutrients in large river basins and reveal its impact on the routing of nutrients throughout the critical zone. Such an informative approach warrants extension to other settings and other isotope systems in order to further reveal the role played by Life on chemical weathering process.





## 6 Appendices

**Appendix A: Identification of dissolved Ba sources in the Amazon Basin rivers**

Spatial variation in dissolved Ba abundance and isotope composition throughout the Amazon Basin could be due to source effects. For this reason, it is necessary to appraise the relative contribution of atmospheric deposition and of release from different rock sources to the Ba river dissolved load.

Rain water is most likely not a significant source of Ba given the low concentration of dissolved Ba in surface seawater (a common source of solutes to rainwater) compared to that of dissolved Ba in rivers (Gaillardet et al., 2003). It is possible to further quantify the contribution of rain-derived Ba following:

$$[Ba]^*_{riv} = [Ba]_{riv} - [Cl]_{atm} \times \left(\frac{Ba}{Cl}\right)_{atm} \tag{A1}$$

where $[Ba]^*_{riv}$ is the dissolved concentration of Ba in the river corrected from rain inputs, $[Ba]_{riv}$ the measured concentration of Ba of the river, $[Cl]_{atm}$ the concentration of dissolved Cl derived from the atmosphere following Gaillardet et al. (1997), and $(Ba/Cl)_{atm}$ the seawater Ba/Cl ratio $\approx 2\text{x}10^{-7}$. Following this calculation, Ba derived from rain inputs is less than 1% of the total dissolved Ba for all rivers analyzed here.

Carbonate rocks formed in ancient marine environments are one of the most important sources of dissolved alkali-earth elements to rivers. However, the abundance of Ba in modern oceans is controlled by barite formation, rather than by carbonate precipitation (Dehairs et al., 1980). None of the Ca*/Na* *vs.* Ba*/Na* and $\delta^{138}$Ba *vs.* Ca*/Na* plots (Fig. S6a,b) show a correlation that could indicate mixing between a silicate and a carbonate weathering end member as a control on dissolved Ba abundance. We can assess the proportion of river dissolved Ba derived from carbonate weathering using the $\alpha^{carbonate}_{Ca}$ (proportion of riverine dissolved Ca derived from carbonate weathering) from Dellinger et al. (2015b) using:

$$[Ba]^{**}_{riv} = [Ba]^*_{riv} - [Ca]^*_{riv} * \alpha^{carb}_{Ca} \times \left(\frac{Ba}{Ca}\right)_{carb} \tag{A2}$$

where $[Ba]^{**}_{riv}$ is the dissolved concentration of Ba corrected from both rain and carbonate inputs; $[Ca]_{riv}$ is the measured concentration of dissolved Ca in the river; and $(Ba/Ca)_{carb}$ the Ba*/Ca* ratio of rivers draining only carbonate rocks, corrected from rain inputs, equal to $5\text{x}10^{-5}$ (median of n = 12 measurements; Fig. S1). The highest proportion of dissolved Ba derived from carbonate rocks was found for the Madre de Dios river (about 10%), whereas carbonate weathering inputs to the Ba dissolved load for other tributaries do not exceed 5%.

In aqueous solutions, Ba shows a high affinity for sulfate ions, which leads to the formation of barite ($BaSO_4$) in oceans and marine sediments. In particular, pre-concentration of Ba and $SO_4$ induced by organic matter decay in micro-environments leads to the precipitation of $BaSO_4$ (Griffith and Paytan, 2012). It has been also shown that the weathering of organic-rich sedimentary rocks such as black shales can be an important input of dissolved Ba to rivers (Dalai et al., 2002) which could be due to presence of barite in these rocks (Arndt et al., 2009). In addition, as an alkali-earth element, Ba could be contained in trace but significant amounts in gypsum, which could also contribute to the river dissolved Ba budget. To test for both





the influence gypsum and barite dissolution, SO$_4$*/Na* *vs.* Ba*/Na* and $\delta^{138}Ba$ *vs.* SO$_4$*/Na* are plotted in Fig. S6c,d. No correlation can be observed between these parameters, implying that neither gypsum nor barite represent significant sources of river dissolved Ba in the Amazon Basin. To conclude, dissolved Ba in the Amazon Basin is almost exclusively derived from silicate rocks, and in our analysis we use our estimates of $[Ba]^*$ as reflecting silicate-derived inputs.

**Appendix B: Parameters and assumption of the Ba isotope mass balance model**

The estimation of $f_{bio}^{Ba}$ from eq. (8) requires the estimation of several parameters, which we review hereafter.

**Source rock Ba abundance ($(Ba/Na)_0$)**

In order to compute the values of $f_{diss}^{Ba}$, the source ratio $(Ba/Na)_0$ first needs to be constrained. In the case of lithologically mixed basins, it can be calculated as:

$$(Ba/Na)_0 = \sum_{i=1} \times \alpha_i^{Na} \times \left(\frac{Ba}{Na}\right)_0^i \tag{B1}$$

with $\alpha_i^{Na}$ the proportion of Na and $\left(\frac{Ba}{Na}\right)_0^i$ the Ba/Na ratio of the rock, and where $i$ = shales, andesites, and plutonic rocks, as carbonate and evaporite rocks do not significantly influence the dissolved Ba and Na budgets, as shown above. The sources of dissolved Na within the Amazon catchment have been identified by Dellinger et al. (2015b). Values and uncertainties for each end member are reported in Fig. S1.

**Source rock Ba isotope composition ($\delta^{138}Ba_{rock}$)**

The value of $\delta_{rock}^{Ba}$ to be used in eq. (8) depends on the relative contribution of each source rock type to the Ba dissolved flux. Three main silicate rocks have to be considered as potential sources of Ba for the Amazon basin: shales, plutonic igneous rocks, and andesites. Using the relative contribution of Na ($\alpha_i^{Na}$) calculated by Dellinger et al. (2015b), we can estimate the relative Ba contribution from each rock following:

$$\alpha_i^{Ba} = \frac{(Ba/Na)_i}{(Ba/Na)_0} \times \alpha_i^{Na} \tag{B2}$$

with *(Ba/Na)$_i$* the average value of the rock end-member $i$; and *(Ba/Na)$_0$* the inferred source rock Ba/Na ratio for each river (see above). The average $\delta_{rock}^{Ba}$ value can then be calculated as:

$$\delta_{rock}^{Ba} = \sum_{i=1} \alpha_{rocki}^{Ba} \times \delta_{rocki}^{Ba} \tag{B3}$$

with $\delta_{rock}^{Ba}$ the Ba isotope composition of each source rock $i$. We use our own measurements for shales ($\delta_{rockshales}^{Ba}$ = -0.02±0.04‰; n = 4) and data from literature (Van Zuilen et al., 2016b; Charbonnier et al., 2018; Nan et al., 2018) for andesites
($\delta_{rockandesite}^{Ba}$ = 0.07±0.02‰; n = 7) and plutonic rocks ($\delta_{rockplutonic}^{Ba}$ = 0.00±0.04‰; n = 71).

**Ba isotope fractionation factor for the formation of secondary weathering products ($\Delta_{sec-diss}^{Ba}$)**





As discussed above in section 4.1, secondary phases, *e.g.* clays or oxy-hydroxides, can at least partially drive the isotope composition of dissolved Ba in rivers (Gong et al., 2019). No experimental study on Ba isotope fractionation during precipitation of secondary weathering phases has been carried out to date, except for sorption on silica hydrogel which did not result in a strong Ba isotope fractionation (Van Zuilen et al., 2016a). Nevertheless, we can try to assess a Ba isotope fractionation factor during formation of clays and oxides using our own measurements and previously published data. Marine clays do not show large variations in $\delta^{138}$Ba, with an isotope composition slightly depleted (by $\sim$ -0.05 ‰) in heavy isotopes with respect to igneous and sedimentary rocks (Bridgestock et al., 2018; Charbonnier et al., 2018). Barium isotope fractionation during formation of secondary weathering products can also be estimated from data on soils. The difference between soil water and bulk solid soil Ba isotope composition over a latosol profile in China display ranges from -0.01‰ to -0.18‰ (average value of $\approx$ -0.09‰), likely induced by the difference in mineralogical composition, and by the varying respective role of sorption and precipitation over the profile (Gong et al., 2019). The large river approach undertaken here likely mitigates such effects of mineralogical heterogeneity, allowing for considering one "lumped" isotope fractionation factor (Bouchez et al., 2013). Our own measurement of Ba isotopes in sediments from the Amazon rivers show the same slight depletion in heavy isotopes compared to shales from the Amazon Basin (this study), or to published data on andesites (Charbonnier et al., 2018; Van Zuilen et al., 2016b), with $\delta^{138}$Ba values ranging from -0.05 to -0.10 ‰, consistent with the estimates reached above. Therefore, taking into account the above constraints and the associated uncertainty, we estimate that $\Delta^{Ba}_{sec-diss}$ = - 0.09±0.05‰.

**Ba isotope fractionation factor during biological uptake ($\Delta^{Ba}_{bio-diss}$)**

Bullen and Chadwick (2016) showed that the isotopic composition of dissolved Ba in the critical zone is strongly impacted by biological processes. Taking the average of the Ba isotope composition of plants and the composition of corresponding rocks (assumed to be representative of the "initial" Ba released in solution after rock dissolution) reported by Bullen and Chadwick (2016) we estimate a $\Delta^{Ba}_{bio-diss}$ = -0.57±0.05‰. We note that although this parameter is poorly constrained, more negative value of this parameter and its position in the denominator of eq. (8) results in the fact that gives the "minimum" of the Ba taken up by the biota.

**Discussing the steady-state assumption**

Eq. (7) assumes that the Ba-bearing materials exported by rivers (excluding primary minerals, *i.e.* dissolved Ba, organics, and secondary weathering phases) correspond to all Ba derived from rock dissolution, thereby achieving Ba steady-state for all critical zone compartments. In such case, the net formation of each secondary phase can be envisaged to be compensated for by an erosion flux, as proposed in Bouchez et al. (2013).

A first "non-steady state scenario" occurs if the "organic reservoir" (biota + soil solid organics) simply grows because litter fall and erosion of organic matter are outpaced by biological uptake. In such a scenario, this would not change the isotopic composition of the dissolved Ba exported by rivers. Therefore, in this case, if the strict steady state assumption is relaxed for the organic (biota plus soil organic matter) reservoir, the steady state isotope mass balance of eq. (7) would still be valid.

Another possibility for non-steady state dynamics is the scenario where litter remineralization releases Ba to waters faster than Ba is taken up by vegetation (implying that the organic reservoir decreases in size), in which case the steady state assumption of the dissolved Ba reservoir is not met, compromising the use of eq. (7) above which derives from a steady-state





framework (Bouchez et al., 2013). We first note that negative $f_{bio}^{Ba}$ values, which should result from such cases where remineralization is outpaced by uptake, are only observed for three rivers from our dataset. Second, if $f_{bio}^{Ba} < 0$, over sufficiently short time scales the isotope composition of dissolved Ba can then rather be interpreted as a mixture between i) a "rock-derived"

end-member, possibly affected by the formation of secondary phases of weathering and ii) an organic-derived end-member.

**Appendix C: River mass budget assumption and derivation**

Here we present in details the different assumption and derivation required to performed river mass budget in section 4.5:

Quantitatively speaking, for an "equilibrated" mass budget, the amount of Ba delivered to the Earth surface by rock uplift and conversion into critical zone material, should be equal to the sum of Ba dissolved ($F_{diss}^{Ba}$) and solid ($F_{sed}^{Ba}$) exports by rivers:

$$F_{diss}^{Ba} + F_{sed}^{Ba} = D \times [Ba]_{rock} \tag{C1}$$

Using term $D \times [Ba]_{rock}$ (with $[Ba]_{rock}$ the abundance of Ba in the source rock) in eq. C1 is equivalent to assuming that the denudation rate $D$ is a proxy for the supply of rock material to the Earth surface. We acknowledge that $D$ can be seen as an imperfect proxy for the supply of rock material to the Earth surface because its validity requires that the export of material by the river is equal to the supply of rock at depth, hence that some form of steady state exists for the regolith covering the Earth

surface. However, we take the fact that in the Amazon the Li river mass budget (using eq. (C1)) is at equilibrium (Dellinger et al., 2015a) as a good indicator that this approach is reliable. We also note that this estimate of Ba supply to the Earth surface neglects inputs from rock-derived nutrients from global dust (Abouchami et al., 2013; Moran-Zuloaga et al., 2018), which can be important in the Amazon lowlands (see discussion in section 4.5).

The same equation can be written for Th, which is an insoluble element with a magmatic compatibility close to that of Ba

(Gaillardet et al., 1999a), thus minimizing the influence of the crustal composition when using rock Ba/Th ratios (see below):

$$F_{sed}^{Th} = D \times [Th]_{rock} \tag{C2}$$

with $[Th]_{rock}$ the abundance of Th in the source rock and $F_{sed}^{Th}$ the flux of Th transported as solids (*i.e.* as sediments) by the river. Then dividing eq. (C2) by eq. (C1) side by side:

$$\left(\frac{Ba}{Th}\right)_{rock} = \frac{F_{diss}^{Ba} + F_{sed}^{Ba}}{F_{sed}^{Th}} \tag{C3}$$





Then dividing the Ba/Th ratio of river sediments $(Ba/Th)_{sed}$ (Table S.5) by the two sides of eq. (C3), and noting that

$$\left(\frac{Ba}{Th}\right)_{sed} = \frac{F_{sed}^{Ba}}{F_{sed}^{Th}}$$

$$\frac{\left(\frac{Ba}{Th}\right)_{sed}}{\left(\frac{Ba}{Th}\right)_{rock}} = \left(\frac{Ba}{Th}\right)_{n} = \frac{F_{sed}^{Ba}}{F_{sed}^{Ba} + F_{diss}^{Ba}} \tag{C4}$$

The term $(Ba/Th)_{n}$ in eq. (C4) correspond to the inverse of the $\alpha$-depletion factor of soluble elements in river sediments, introduced by Gaillardet et al. (1999a); and is conceptually equivalent to $1 + \tau_{Ba,Th}$ where $\tau_{Ba,Th}$ is the mass transfer co-

efficient developed for quantify the depletion in soluble elements in soils (Brimhall and Dietrich, 1987). Finally, given the definition of $w^{Ba}$ (eq. (3)), called hereafter "$w_{fluxes}^{Ba}$" (for better distinction with "$w_{iso}^{Ba}$" calculated below), an equilibrated river mass budget should thus result in:

$$(Ba/Th)_{n} + w_{fluxes}^{Ba} = 1 \tag{C5}$$

Both parameters should in principle take values between 0 and 1, and we note that in our dataset, as Ba is always depleted

over Th in the measured river sediment material compared to rocks, $(Ba/Th)_{n}$ is between 0 and 1. Regarding $w_{fluxes}^{Ba}$, it is important to note that eq. (3) can be used based on river sediment gauging (yielding the averaged $[spm]$ as in (3), or through:

$$w_{fluxes}^{Ba} = \frac{Q \times [Ba]_{diss}}{D \times [Ba]_{rock}} \tag{C6}$$

Using (3), $w_{fluxes}^{Ba}$ is by construction between 0 and 1, whereas if eq. (C6) is used, it is possible that $w_{fluxes}^{Ba}$ takes values higher than 1 because the numerator and the denominator of the right-hand side of eq. C6 are evaluated in independent ways,

and using metrics reflecting different time scales of critical zone dynamics. However, in our dataset, regardless of the way $w_{fluxes}^{Ba}$ is calculated all values calculated for $w_{fluxes}^{Ba}$ in the Amazon Basin are between 0 and 1.

$(Ba/Th)_{rock}$ in $(Ba/Th)_{n}$ is estimated for each river using the rock contributions given by Dellinger et al. (2017), and the Ba/Th ratios of individual rock types (equal to 48 for igneous plutonic and shale rocks - which have a similar Ba/Th ratio - and of 140 for andesites; Table S.6).





## Appendix D: Calculation of dimensional, catchment-scale Ba biological uptake fluxes

The isotope mass balance model presented in section 4.7 allows to estimate $f_{bio}^{Ba}$, which a relative, hence non-dimensional flux of Ba biological uptake, with respect to the flux of Ba release from rock partial dissolution. In section 4.3, our analysis requires a dimensional values equivalent for $f_{bio}^{Ba}$, but for other rock-derived, major nutrients.

Recalling the definition of $f_{bio}^{Ba}$:

$$f_{bio}^{Ba} = \frac{F_{bio}^{Ba}}{F_0^{Ba}} \tag{D1}$$

and applying the same definition for another element X:

$$f_{bio}^{X} = \frac{F_{bio}^{X}}{F_0^{X}} \tag{D2}$$

Now recalling the definition of $f_{diss}^{Ba}$:

$$f_{diss}^{Ba} = \frac{F_{diss}^{Ba}}{F_0^{Ba}} \tag{D3}$$

and applying the same definition for another element X :

$$f_{diss}^{X} = \frac{F_{diss}^{X}}{F_0^{X}} \tag{D4}$$

we can re-arrange these equations and obtain:

$$\frac{f_{bio}^{X}}{f_{diss}^{Ba}} = \frac{F_{bio}^{Ba}}{F_{diss}^{Ba}} \tag{D5}$$

such that the net, dimensional flux of *X* biological uptake is:

$$F_{bio}^{X} = Q \times [Ba]_{diss} \times \frac{f_{bio}^{Ba}}{f_{diss}^{Ba}} \times \left( \frac{X}{Ba} \right)_{bio} \tag{D6}$$





*Author contributions.* QC performed new analytical work, interpreted data, and wrote text. JB and JG designed the study, conducted field work and performed previous analytical work used in the present contribution, interpreted data, and wrote text. EG performed the analysis of remote sensing data and wrote text.

*Competing interests.* The authors declare that they have no conflict of interest.

*Data availability.* Supplementary data tables are available in the online version of the paper.

*Acknowledgements.* The authors are grateful to Pascale Louvat, Thibaud Sontag, Jessica Dallas, Caroline Gorge, and Pierre Burckel for analytical support. Nicole Fernandez is thanked for English corrections. Geochemical analyses presented in this study were enabled by the IPGP multidisciplinary program PARI and by the Region Île-de-France SESAME Grant no. 12015908, and by the grant "Émergences" awarded by the City of Paris to Julien Bouchez.





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
