# Peer review of "Barium stable isotopes as a fingerprint of biological cycling in the Amazon River Basin"

_Biogeosciences, 2020_

## Referee Comment (RC1) · Anonymous Referee #1 · 12 May 2020

Manuscript summary: In the manuscript entitled "Barium stable isotopes as a fingerprint of biological cycling in the Amazon River Basin" Quentin Charbonnier and co-authors measured barium stable isotopes on dissolved loads and suspended sediments of the Amazon River and its tributaries. They found that the dissolved load is isotopically different to parent silicate rock and attribute this difference to both the formation of secondary weathering products and Ba uptake by biota. Applying an isotope mass balance model, they find that Ba is strongly cycled through biota. Moreover, including rates of denudation the authors find a deficit in the Ba export, which they interpret to be of organic matter that is either exported in particulate form or accumulates in the study area. Finally, by correcting the river derived silicate weathering flux from the biological imprint they conclude that the CO2 consumption by chemical weathering

is by about 20% underestimated.

Main Comments: This manuscript is well-organised and well-written. Given the presentation of in total 35 equations the authors have done a great job in writing, which allows the reader to easily follow the story. However, even though the English in general is fine the manuscript would benefit from proofreading by native speakers. Below in the line by line comments I waive from commenting on language issues. The presentation of the data and data quality is very good. The story is mostly scientifically sound. I have only three major concerns namely i) time scale issues and Ba uptake fluxes by biota ii) section 4.4, and iii) section 4.7.

Regarding i) even though I like the geochemical mass balance approach of how to estimate absolute fluxes in this manuscript I am missing a discussion on time scale issues to allow the reader to judge the applicability of the findings. This is because total denudation rates from cosmogenic nuclides integrating over 10ˆ3 to 10ˆ5 years and sediment gauging data integrating over some tens of years are used in the metrics of this manuscript. What is more, the authors estimate the Ba uptake flux of plants independent from direct measures by their mass balance approach. Principally, this is fine to me. But, given that GPP data is available I also would like to see a comparison of how their FˆBa_bio compares to a GPP-based Ba uptake flux by plants (e.g. using Ba concentrations in plants and GPP data).

Regarding ii) I am surprised about the trends shown for GPP vs. W/D and TER vs. W/D. Keeping in mind that my understand is that GPP is a proxy for total biomass production plus respiration (actually I think NPP would be a more appropriate metric for the purpose of this manuscript), TER is a proxy for nutrient recycling and W/D is a proxy for the weathering intensity I am surprised that both recycling and total biomass production increase with increasing weathering intensity. To my understanding increasing recycling requires energy and hence goes on the costs of biomass production unless there is a counterbalancing effect. This counterbalancing effect could be due to a gradient in MAP and/or MAT because biomass production primarily depends on energy,

temperature, and water & nutrient availability. Thus, I recommend providing MAP and MAT in the study area section. On the other hand I am wondering that recycling increases with increasing W/D. Increasing W/D can either be understood as decrease in nutrient supply (because soils are heavily depleted, high W, leading to the need of recycling) or increase in nutrient supply (because soils are replenished by less weathered minerals, high E, no need for recycling). But recycling must not only be a nutrition strategy to cope with low nutrient supply through weathering or atmospheric deposition as recycling is also considered as the major nutrition mechanism (e.g. fast turnover of organic matter minimizes nutrient loss from the system). Importantly, there seems also to be a misunderstanding on the organic and geogenic pathway presented in Uhlig and von Blanckenburg 2019. This is, because both pathways operate simultaneously but in terms of fluxes (and on the short timescale) the organic pathway is way more important than the geogenic pathway which becomes essential on the centennial to millennial timescale. Given these complexities in ecosystem nutrition the authors are not putting their findings well in context of ecosystem nutrition. Thus, the authors somehow over-interpret their findings in section 4.4 because of the generalization made and because the field of ecosystem nutrition is widely ignored (e.g. lacking in the introduction). For this reason, I either recommend down tuning of the conclusion or discussing their relationships in the context of ecosystem nutrition (the latter would likely bias the focus of the manuscript). Moreover, that Ba is so strongly cycled through biota is surprising as to my knowledge Ba is neither essential to plants nor considered as a micronutrient or even as a plant beneficial element, following definitions by Marschner (2011) . Thus, the strong cycling of Ba through plants in the giant ecosystem of the Amazon may warrant more attention in the manuscript.

Regarding iii) I basically did not understand why the silicate weathering flux should be corrected for the imprint of the biosphere. This is because mineral nutrients, which were cycled through biota are mainly (apart from atmospheric inputs) of geogenic origin, hence released by chemical weathering then cycled through the biosphere and ultimately appear with some delay in the river. I may misunderstood section 4.7, thus

please revise this section for more clarity.

In consideration of these main comments I suggest major revisions.

Text: Line by line detailed comments:

L.4 delete "measured" as it is clear that the data was measured

L.14 please specify through what Ba is cycling exactly (e.g. biota, secondary minerals)

L.16 see comment L.14

L.21 To my understanding "minor nutrient" is not a well-defined term for nutrients. Nutrients are either called "essential" elements for plant growth or grouped into the categories "macronutrient", "micronutrient" and "plant beneficial element". Following Marschner (2011) Ba belongs to none of these groups.

L.24 for completeness it would be nice to also tell the reader the "yet reported value"

L.32 "making up our water quality" is out of place as the manuscript is not about water quality

L.48 The reference Napieralski et al. 2019 maybe useful here, but I do not think it is the most appropriate one for the general information made here.

L.59 The role Ba plays in plants should be briefly introduced. I also do not think that the Viers et al. 2005 reference is the most appropriate one.

L.88 should read "drainage area"

L.90 Mean annual temperature (MAT) and mean annual precipitation (MAP) should be provided as they may be important for section 4.4

L.96 Providing data on erosion rates (e.g. ranges) would help the reader to assess how high or low the rates really are.

L.105 see comment L.96

[Figure]

L.117 Please remove "are from . . . repository" as it is not important to tell where samples are stored

L.122 It reads that cation concentrations were measured by ion chromatography. Please revise if incorrect.

L.140 Please provide the typical sample weight used for digestion.

L.145 Which matrix elements were eluted with 2.5 M HCl and how many mL of 2.5 M HCl and 6M HCl were used?

L.146 What is meant by "carried out sequentially twice"? Needs rewording

L.152 "(up to 300 ng)" can be removed as this information is included in the range 250-300 ng L.153 "coupled to" should be replace by "using"

L.154 I am wondering why so many cups were used. Please explain why each of the measured isotopes were needed for data evaluation. The ones which are measured but not used for data evaluation must not be mentioned.

L.163 I am wondering why the delta 138Ba was calculated from delta 137Ba and not from the measured isotope ratios 138Ba/134Ba (as is done for delta 137Ba by using measured 137Ba/134Ba).

L.175 To allow the reader to evaluate how the data on reference materials measured in this study compare to literature values please also provide the previously published delta values.

L.253 To conclude that Ba behaves like a nutrient solely from the correlation of K/Na and Ba/Na is not very reasonable. Provided that stream water integrates over all critical zone processes this correlation could have several reasons. Importantly, K is much more soluble when rainwater passes through the canopy (throughfall) than other mineral nutrients. Thus, I am wondering whether Ba is also as mobile as K from the canopy (apparently not as the authors find that Ba is exported in particulate organic form or

accumulates in the Amazon biomass).

L.272 I am wondering whether the term "model" is appropriate here as later in the manuscript (e.g. L.342) the authors call it "mass balance approach" which I think is the more appropriate term. L.330-335 Given that D is E+W this section would benefit from saying when E (or W) is high or low.

L.340 Please split the sentence into two sentences. The first should be on Ba, the second on Li. Also please say that Li has to date not been shown to fractionate during plant uptake.

L.343 I do not really see "correlations" in figure 5. Thus, either revise the wording or show a figure that illustrates correlations.

L.346 Another explanation could be that Ba is in principle less soluble than other elements (as stated earlier in the manuscript).

L.348 To me $f\hat{}Ba\_bio$ is not "fairly constant". It simply does not show a trend but the $f\hat{}Ba\_bio$ data is too different to say that they are constant.

L.349 see comment L.343

L.351 Please use another word for "automatic". I do not understand what the authors want to say.

L.355 typo, should read "eq. (10)"

L.357 Please remove "that we convert in . . ." as this conversion is trivial.

L.384 "rock supply rate" means "regolith production"? Please clarify.

L.399 Beginning from here I got confused and I think one of the reasons is that the authors should be more precise when presenting low or high W/D ratios. This means the reader needs to know that high or low W/D ratios are due to high or low W or E (D= W+E). This differentiation is quite important in order to understand why both

GPP and TER have a positive relation to W/D. As explained in the main comments it is somehow surprising that GPP (nutrient uptake) and TER (nutrient recycling) increase with increasing W/D.

L.410 To my understand TER does not say anything about the size of biota, if any, GPP would do so. Please revise.

L.413 As briefly explained in the main comments there seems to be a misunderstanding on the organic and geogenic pathway presented in Uhlig and von Blanckenburg 2019.

L.422 "made available through denudation" Please specify whether E or W in denudation is meant here, as both can have different effects on plant nutrition.

L.424 Please specify which elements are close to Ba in terms of their nutritional role.

L.426ff I doubt that larger pools of bio-available Ba "enhance" Ba uptake. First, the bio-available pool must be continuously replaced over time and second given that Ba is not a nutrient with high bio-available Ba pools more Ba might be taken up but uptake not enhanced. Hence, please revise the wording. I am also wondering why Ba should first be incorporated and then taken up from secondary weathering products. My understanding is that is less energetic to plants to take up dissolved nutrients. As Ba, which was incorporated into secondary products was already dissolved, I would assume plants should have utilised Ba before its incorporation into secondary products.

L.433 Please check whether water residence time in the critical zone is longer at steeper slopes. To me the opposite makes sense.

L.473 Please specify what is meant by an "intensive" metric.

L.478 To assess whether there is really a difference between $\hat{w}Ba\_fluxes$ and $(Ba/Th)n$ uncertainties would be helpful which are missing in figure 8a. Also, provided that $\hat{w}Ba\_fluxes$ relies on Q and D (equation C6) integrating over much different timescales I am wondering whether the differences in figure 8a could also be explained by time scale issues.

L.534 fˆBa_miss is a relative flux and should be named as such to not confuse the reader as the reader is anyhow busy in differentiating when absolute and when relative fluxes are presented.

L.536 should read "with the relative amount" as fˆBa_bio is a relative flux.

L.543 Please let the reader know the time scale.

L.552ff I am surprised about the groundwater discussion as to my understanding rivers, particularly at baseflow, are mainly fed by groundwater. Or do the authors mean an over regional groundwater aquifer contributing to streamflow?

L.585 Either the C in the left-hand term (FˆC_miss) should be removed or *[C]_org added to the right-hand term in equation 15. Otherwise the equation is wrong.

L.586 see comment in L.585

L.591 please add "particulate" to "organic carbon"

L.609ff The section heading reads that it is valid for all mineral nutrients. Please add "Ba" in order to avoid a generalization which may not hold for other mineral nutrients. Figure 11 could be called 11a and 11b which would allow the authors to cross reference each scenario mentioned in section 4.6 directly to the scenarios shown in figure 11. Please see again major comments (ii) for revision.

L.615 "new" Ba is a formulation from Cleveland et al. (2013) (new P), thus please cite.

L.628 Please see major comments (iii). Moreover, I do not see a clear trend in figure 12, thus high or low R_sil+bio/sil should be treated with caution.

L.636 "the uptake flux of this element" is misleading because [X]_bio represents a concentration in equation 17.

L.702 better use "assess" instead of "appraise"

L.707 "atm" in "(Ba/Cl)_atm" should be replaced by "seawater" as according to the text

it represents the seawater ratio?

Figure 1 Axis caption "latitude, longitude" is missing. Also, the left-hand part of the altitude scale can actually be removed as information on the altitude is more relevant for the sample locations.

Figure 2 Please add the per mil sign next to delta 138Ba

Figure 6 I am wondering why "bio" and "sec" show trends for absolute fluxes when plotted against D or W/D in Figure 6 but not for relative fluxes in Figure 5. Provided that in this manuscript so many equations, ratios and ratio over ratios are presented please explain this difference. Regarding style, please frame the legend. Instead of using "+" in this and the other figures please write "increasing" or "decreasing" as the plus is confusing. Also, please remove the arrow and "Biological uptake of rock-derived nutrients" as FˆBa_bio only shows Ba and no other elements and as this information does not provide additional information that the axis label and legend itself.

Figure 7 In order to better assess whether the trends in this figure are real I would like to see uncertainties. The arrows and text next to the y-axis of panel a and b can be removed as they do not provide additional information. In panel d, I believe there is a misunderstanding regarding the "organic" and "geogenic" nutrient pathway as in eroding systems both are present. The main pathway is the organic one (most important on time scale of up to a few years only) and the geogenic one is minor in terms of fluxes but most important on the centennial to millennial timescale. Thus, it makes no sense to put FˆBa_bio in dependence of both pathways in this figure.

Figure 8 In the figure caption it says "Uncertainties on wˆBa_fluxes" but in panel 8a no uncertainties are shown. Is this correct or are the uncertainties smaller than symbol size. This needs clarification. Regarding uncertainties I also noticed that the error bars are differently long towards positive and negative directions. Please check.

Figure 9 The arrows next to the x-axis and y-axis do not provide additional information

[Figure]

and should be removed. Latest with this figure I would like to see a comparison of the FˆBa_bio with a more direct estimate of the Ba uptake flux, e.g. based on GPP and Ba concentrations in plants.

Figure 11 Please name both panels (e.g. "a" and "b"), which allows the authors to cross reference the figure more detailed in the main text (section 4.6). Also in the figure caption tell the meaning of the arrow sizes.

References mentioned in this review

Cleveland, C. C., B. Z. Houlton, W. K. Smith, A. R. Marklein, S. C. Reed, W. Parton, S. J. Del Grosso, and S. W. Running. 2013. "Patterns of New versus Recycled Primary Production in the Terrestrial Biosphere." Proceedings of the National Academy of Sciences 110(31):12733–37. Marschner, P. 2011. Marschner's Mineral Nutrition of Higher Plants.
* * *

---

## Referee Comment (RC2) · Anonymous Referee #2 · 16 Jun 2020

The manuscript "Barium stable isotopes as a fingerprint of biological cycling in the Amazon River Basin" by Charbonnier et al. describes the use of stable Ba isotopes as proxy for estimating the role of the biota on Ba fluxes in large catchment areas. Based on the approach of using stable isotope proxies (e.g., Li) to calculate river mass budgets, Charbonnier et al. found that part of the Ba dissolved from the bedrocks in the catchment is missing. Following several lines of evidence, based on (isotope) mass balance calculations, correlations with ecosystem dynamics and river mass budget, they concluded that biological cycling in the critical zone is responsible for the missing Ba component. Further implications of their findings are that earlier calculations of the $CO_2$ consumption of the catchment, based on river cation loads, might be considerably underestimated.

General comments:

The manuscript is well written and organized and a reader can follow the authors' argumentation. The English of the second version of the manuscript benefited greatly from proof reading after the access reviews. I encourage the authors to also check the supplement for English grammar and style. I recommend publication of the manuscript in *Biogeosciences* after moderate revisions. My comments, which I hope the authors will find useful and constructive, are listed below.

Specific comments:

The main conclusion of the study is that a considerable amount of Ba dissolved from the bedrock and transported by the rivers is taken up and stored by biota. I was wondering how exactly plants and/or (micro)organisms utilize Ba. To my knowledge, Ba is not considered an important nutrient. The authors even state that Ba could be a limiting factor for biota growth (page 19, line 424). This has to be further discussed.

Did the authors propagate uncertainties of single parameters in their models? Most figures do not have error bars and it is thus difficult to assess whether apparent trends are real or not within uncertainty. Furthermore, I am missing estimations on the uncertainties of, for instance, the 20% underestimated $CO_2$ consumption, the main impact this study might have.

For the isotope mass balance in section 4.2 the authors assume congruent dissolution of the bedrock, i.e., no Ba isotope fractionation. Assuming this assumption is wrong, how large would be the impact of isotope fractionation during rock dissolution on the model output? Would it be negligible?

In section 4.4 the authors describe an apparent trend in biological Ba cycling with ecosystem dynamics (Fig. 7c,d). As the figure is now, I fail to see the trend. The only obvious is that the Madeira tributaries have lower GPP and TER values than the rest. However, there is a discrepancy between GPP data in Fig. 7a and 7c. Also, error bars are missing.

In section 4.7, I do not agree with the authors' interpretation that $R_{(sil+bio)/sil}$ increase with very low W/D, based on Fig. 12. The argumentation is apparently based on one data point. Also, this figure lacks error bars.

Appendix B: The authors estimated the Ba isotope fractionation between dissolved Ba and Ba taken up by biota, admitting that it is poorly constrained. Yet, they state the fractionation with a fairly high precision of ±0.05 ‰. How reliable is the estimated fractionation?

The authors made a great effort in computing and quantifying data and parameters. However, not all derivations of equations can be followed easily. For instance, I failed to understand how equations C5, D5 and D6 are derived given the provided information.

Page 3, line 59: $^{130}$Ba is a primordial nuclide and can be considered stable under geochemical aspects.

Page 5, line 134: What are plutonic rocks in this case? What is their lithology?

Page 10, line 214: Please define * in the main text, not only in the figure caption and supplement.

Page 19, line 433: Why is the residence time of water longer along steeper slopes?

Page 29, line 654: I could not find any data/figure supporting the argument that mainly K weathering flues are influenced by biological cycling. If they are to be found, e.g., in the supplement, please refer to it. Otherwise data have to be provided.

Page 30, line 669: Please quantify this significant uncertainty!

Page 31, line 672: […] to be source mainly from silicate rocks […]

Page 33, line 754: Charbonnier et al. (2018) is a review paper. When literature data are used, please cite the original publications (also later in that appendix).

Technical comments:

Fig. 7: GPP data are different in panel a) and c)!

Fig. S2: Are the error bars correct? They show approximately ±0.15 ‰ on $\delta^{138/134}$Ba. Long-term precision for BaBe27 and JB-2 is however given as ±0.08 ‰.

---

## Author Comment (AC3) · 3 Jul 2020

**Final answer to the reviewers**

We first want to acknowledge both reviewers for their insightful comments. We are glad to see that the originality of the method presented in our manuscript (*i.e.* the use metal stable isotope composition of river material associated to mass balance equations, with direct comparison of ecosystem productivity) has been appreciated.

Nonetheless, bridging the gaps between Earth surface geochemistry and ecology is not an easy task. Indeed, reviewers highlight weaknesses in our interpretation of ecosystem nutrition pathways, leading to some overstatement and wrong generalization on the behavior of rock-derived nutrients. Also, in our manuscript the focus on "weathering" parameters (such as weathering intensity; W/D) to explain ecosystem nutrition was too strong.

According to the reviewers' comments and as explained in the separate replies, we will take the opportunity through this review to improve our manuscript. In particular:

- Qualifying Ba as a "nutrient" is wrong as Ba does not have any physiological role in plants. Thus, we would define it as having a "nutrient-like behavior" instead. From here, we will state as a working hypothesis the fact that the behavior of Ba only broadly reflects that of rock-derived nutrients (*e.g.* Ca, Mg, P...). More generally, throughout the manuscript generalization of our findings to rock-derived nutrients will be tuned down (revs. 1 and 2).
- The potential role of some important metrics for biological productivity such as precipitation and temperature has been ignored. We will give them a more important place within the discussion (rev. 1).
- The isotope-derived flux of net Ba uptake could be compared directly to a similar estimate from GPP (and the typical Ba concentration in biological material). We think that this is a good idea, although (1) the comparison should be made using NEE = GPP TER, representing the net increase in biomass should be used because (2) it is more informative to compare the concentration of Ba in organic matter required for these two metrics to agree, and then to compare this concentration to independent estimates of Ba concentration plants. We will add a short discussion text accordingly (rev 1).
- A part of our discussion relies on the definition of the nutrient uptake through "geogenic" and "organic" pathways. The way we use these definitions was misleading, and we will clarify this in the next version of the manuscript (rev. 1).
- We will add error bars to the diagrams, where missing (rev. 2).

---

## Author Response (AR1)

Dear Dr. Veldkamp,

We thank you for leaving us the opportunity to improve our manuscript, which has now been thoroughly revised in order to address all reviewers' comments along the following lines:

- First, we added reference to a significant body of literature in the introduction and in our discussion dealing with ecosystem nutrition strategies.

- Second, we removed the term "nutrient" to qualify Ba, as this term was incorrect. We replaced it by "nutrient-like behavior" instead. We also better explain now why Ba can be considered as having a nutrient-like behavior in the manuscript introduction.

- Third, the section on the comparison between ecosystem nutrition strategies and the Ba isotope mass balance have been strongly re-written and now better reflects the current knowledge (and remaining uncertainties) on ecosystem nutrition. We also changed Figure 7 in the related section to emphasize the potential role of climatic parameters (rather than denudation rates and weathering regimes) in driving Ba biological cycling.

- Finally, we added error bars in diagrams wherever they were missing. To do so, new uncertainties had to be calculated for some parameters (*e.g.* the "$F^{Ba}_i$"), and were mostly assessed using Monte Carlo simulations, resulting in non-symmetrical parameter value distributions. Therefore, we had to shift to using the median of these distributions instead of the mean. This leads to slight changes in the reported central estimates of these parameter values and to some extent in modifications in the relationships between parameters in Figures 5, 6, 7, and 9. Nevertheless, these modifications leave our main interpretations unchanged. Because the previous version of Figure 5 (which was a "stacked bar plot") did not allow for a clear representation of error bars, it has been changed into a scatter plots. We also note that the values of the silicate weathering fluxes corrected for biological uptake (see section 4.7 and Table S.8) are not significantly higher (on average twice higher, instead of +20% higher in the previous version, compared to non-corrected silicate weathering fluxes). This differences stems from the fact that negative draws of necessarily positive parameters (for example on the $(X/Ba)_{bio}$ with X = Mg, Ca, K; see eq. 18) during Monte Carlo simulations were discarded. Obviously, these new central estimates for this parameter are accompanied with large uncertainties, and although the point we were making in this section remains the same (namely that it is necessary to take into account the biological storage of cations derived from silicate weathering when estimating silicate weathering fluxes from river data) we tuned down our conclusions and removed the corresponding figure (formerly Fig. 12) basing our analysis only on values reported in a table now.

Please find below the reviewer comments (in black) and our answers from the interactive discussion (in blue) and the manuscript line(s) of the corresponding changes during revision - if any (in green).

Answer to the Reviewer #1.

We thank Reviewer #1 for his/her thorough report on our manuscript. His/her main concerns revolve around three aspects of the manuscript, namely i) potential time scale effects that could explain some of our observations, ii) the relevance of the weathering regime metrics we use in driving ecosystem dynamics and rock-derived nutrient cycling, and iii) our contention that biological uptake of rock-derived nutrient might induce a bias when catchment-scale weathering rates are estimated from river chemistry data. To address the concerns raised by reviewer 1, we plan to significantly rewrite some parts of the manuscript (in particular sections 4.4 and 4.7) during revision, and below we provide clues about how we want to proceed to do so.

This manuscript is well-organised and well-written. Given the presentation of in total 35 equations the authors have done a great job in writing, which allows the reader to easily follow the story. However, even though the English in general is fine the manuscript would benefit from proofreading by native speakers. Below in the line by line comments I waive from commenting on language issues. The presentation of the data and data quality is very good. The story is mostly scientifically sound. I have only three major concerns namely i) time scale issues and Ba uptake fluxes by biota ii) section 4.4, and iii) section 4.7.

We are glad that the reviewer values our approach and appreciates the overall structure and writing of the manuscript. This version was already proof-read before submission by a native (US) English speaker, but before submission of the revised version we will ask again for proof-reading to another native speaker.

Regarding i) even though I like the geochemical mass balance approach of how to estimate absolute fluxes in this manuscript I am missing a discussion on time scale issues to allow the reader to judge the applicability of the findings. This is because total denudation rates from cosmogenic nuclides integrating over $10^3$ to $10^5$ years and sediment gauging data integrating over some tens of years are used in the metrics of this manuscript.

The time scale issue is indeed a recurring problem in geochemical mass balance approaches. We actually see two questions embedded in this particular remark:

(1) The timescale over which erosion is "measured" differs between sediment gauging and cosmogenic nuclides. However, as explained in our manuscript, only a handful of Andean rivers display a significant (by more than a factor of 2) difference in erosion measured by the two methods, most likely due to a recent ($< 10^2$-$10^3$ yrs old) increase in erosion (Wittmann et al., 2011). However, we believe that this potential effect is taken into account in our analysis as when $w^{Ba}_{fluxes}$ (the calculation of which relies on the knowledge of erosion rate) is plotted (figs. 8 and 9) both sediment gauging- and cosmogenic nuclide-derived values are shown. Taking one or the other does not affect our conclusions.

(2) Beyond their effect on erosion rates, time scale might affect other variables we use too, and thus explain some of the differences we observe between parameters. Indeed, the values of all parameters of the river mass budget equations (eqs. 11 and 12, and eqs. C1 to C6) are likely to depend on the timescale over which they are estimated. And in a way, part of our discussion in explaining the differences between the various river mass budget parameters is indeed about time scale effects. For example, when we propose that the export of particulate organic debris could account - at least partially - for the observed disequilibrium in the Ba river mass budget, we assume that such a component of the river flux would be missed because they could be preferentially exported during relatively rare events such as floods. This is typically a time scale issue. Same for the scenario we propose where this particulate organic carbon is currently accumulating in the soils of the basins rather than being exported by rivers - would sampling of rivers last over much longer time scales, it would be very plausible that this organic carbon is exported by erosion and/or dissolution (because it cannot accumulate indefinitely), and the difference would disappear. Of course, there is a daunting number of time scale effects that could shift the values of individual variables of our approach, and result in the observed differences. Some of them have already been explored in previous studies (see *e.g.* Stallard et al., 1995; Gaillardet el al., 1995, 1997; Lemarchand et al. 2006; Bouchez et al., 2013). We do not see how we could cover all these possibilities in our manuscript. However, first during revision where relevant we will emphasize how our current interpretations relate to time scale issues. Second, we

think the fact that the river mass budget of Li across the Amazon Basin rivers is equilibrated (see section 4.5 and fig. S5) suggests that these time scales issues are minor for the "inorganic" part of the rock-derived nutrient cycle (Dellinger et al., 2015). During revision, we will also emphasize how we think that Li isotopes indicate that some of the potential time scale issues do not affect our analysis.

We mentioned the possibility of timescale issues in the river mass budget (lines 486-489) and explained why we can rule out this explanation to our findings (lines 520-526).

What is more, the authors estimate the Ba uptake flux of plants independent from direct measures by their mass balance approach. Principally, this is fine to me. But, given that GPP data is available I also would like to see a comparison of how their FˆBa_bio compares to a GPP-based Ba uptake flux by plants (e.g. using Ba concentrations in plants and GPP data).

First, we would like to emphasize that within our mass balance framework, only the *net* formation of an isotopically-fractionated phase (*i.e.* its formation rate minus its destruction rate) is important for the isotope budget (lines 288 to 296 of our manuscript and Bouchez et al., 2013). In this scope, we believe that NEE = GPP - TER (ecosystem respiration) is a more sensible metric of net formation of organic matter to which we should compare our isotope-derived $F^{Ba}_{bio}$ parameters.

[Figure]

Nevertheless, we acknowledge that the suggestion of the reviewer is a good one for the sake of the discussion. As shown in the attached figure, the uptake of Ba calculated from the isotope budget (eq. 8) is much larger than that based on GPP and NEE, for the Andean tributaries. By contrast, in the main and "dilute" tributaries, where the isotope-derived Ba uptake is lower the GPP-based Ba uptake is much larger. However, at this point we need to emphasize the poor constraints available on the Ba content of organic matter, which most likely entails significant uncertainty in this approach (let alone the uncertainty on NEE, which is calculated from a difference between two large fluxes, GPP and TER, each of those also being affected by significant uncertainty). An alternative to assess the broad reliability of our analysis is to actually estimate the average Ba content of the organic matter formed ("net", *i.e.* remaining from GPP after plant and soil respiration as quantified by TER), through the combination of $F^{Ba}_{bio}$ and NEE following:  $F^{Ba}_{bio}$ / (NEE x $1/[C]_{org}$) = $[Ba]_{org}$.

By doing so, we find that a Ba content around 5 to 10 ppm is necessary in exported / accumulated organic matter for the Andean belt rivers – which remains an acceptable result regarding the Ba content of plants we can extract from our own compilation (18 ± 28 ppm; S.D). However, the calculated Ba content in exported organic matter drops between 1 to 0.1 ppm for the main and "dilute" tributaries, *i.e.* in the lowland areas of the Amazon. This is an interesting piece of result, as such lower predicted organic matter Ba content may come from a preferential release of Ba from litter during intense recycling. This idea is in agreement with the much higher $f^{Ba}_{diss}$ (*e.g.* > 1 for the Rio Negro River) and stronger export of Ba by rivers than theoretically predicted for "dilute" rivers (see "Dilute tributaries" in fig. 8a). This is actually an interesting finding, and our suggestion for the revision is to include a short text (2-3 sentences) to explain this at the beginning of section 4.4, without adding the figures given above with the direct comparison between the isotope-derived $F^{Ba}_{bio}$ and the GPP- or NEE-counterparts.

We added this comparison (lines 645-660).

Regarding ii) I am surprised about the trends shown for GPP vs. W/D and TER vs. W/D. Keeping in mind that my understand is that GPP is a proxy for total biomass production plus respiration (actually I think NPP would be a more appropriate metric for the purpose of this manuscript), TER is a proxy for nutrient recycling and W/D is a proxy for the weathering intensity I am surprised that both recycling and total biomass production increase with increasing weathering intensity.

First, and as explained in our manuscript, most of the ecological parameters used here are actually calculated based on GPP (Tramontana et al., 2016; Jung et al., 2017). As a result,

[Figure]

TER and NPP strongly correlate with GPP (see figure below). Also, we specifically used in the manuscript GPP as a proxy for the size of the biomass, while TER is used as a proxy for remineralization. For these reasons, we plant on keeping the use of GPP and TER in that way.

We kept the used of TER and GPP as said in the answer.

Second, regarding the "surprising" relation between W/D on the one hand and GPP and TER on the other hand, we gather that clarification is needed. In the Amazon Basin, W and D are positively related. Nonetheless, D increases (from the plain to the Andes) much quicker than W. Therefore, the W vs. D relation is that of a diminishing return, resulting in a negative relationship between W/D and D (see figure: this is because at high D a "kinetic limitation" is imposed on mineral dissolution - hence W - due to the short residence time of solids in the critical zone; see for example Ferrier and Kirchner 2008; Gabet and Mudd 2009; Dixon et al., 2012). Therefore, the positive relations

between W/D and GPP or TER should not come as a surprise as the largest ecosystems of tropical forests in the Amazon Basin are found in the plains – where W/D is high.

We clarified the difference between denudation fluxes (W and D) and weathering intensity (lines 368-370).

To my understanding increasing recycling requires energy and hence goes on the costs of biomass production unless there is a counterbalancing effect. This counterbalancing effect could be due to a gradient in MAP and/or MAT because biomass production primarily depends on energy, temperature, and water & nutrient availability. Thus, I recommend providing MAP and MAT in the study area section.

We fully agree with the review that MAP and MAT are important parameters for ecosystem nutrition that were pretty much not discussed in the first version of our manuscript. Therefore, we will include a discussion of the role of these parameters in the appropriate section and add the MAP and MAT of each basin in the table. We will also add panels to an existing figure to showcase the potential role of MAP and MAT, such as in the attached figure, where we note that $F^{Ba}_{miss}$ displays a negative relation with both MAP and MAT. Hence, indeed lower MAP and MAT (in other words, a lesser source of energy required for efficient organic matter remineralization) are associated with a larger missing flux of Ba. Therefore, water and temperature probably modulate the role of weathering regime on nutrient cycling, something that was

not apparent in the previous version of the manuscript. We will significantly amend the text of section 4.4 to reflect this fact.

MAP and MAT have been provided in the geological and geographical setting and are now part of the whole discussion (see sections 4.4 and 4.5).

On the other hand I am wondering that recycling increases with increasing W/D. Increasing W/D can either be understood as decrease in nutrient supply (because soils are heavily depleted, high W, leading to the need of recycling) or increase in nutrient supply (because soils are replenished by less weathered minerals, high E, no need for recycling).

Our answer to this comment echoes the answer given above about the relation between D and W/D, and we think that part of the misunderstanding might lies in a confusion between weathering intensity (W/D) and flux (W). High-W/D settings (typically featuring deep, heavily depleted soils such as those found in the Amazon lowlands) are characterized by low W (and low D, so low E too). So the reviewer is right saying that increasing W/D corresponds to a decrease in nutrient supply from weathering indeed (but this corresponds to low W, actually), but increase in nutrient supply (from weathering, not from recycling of organic matter) is necessarily associated to low W/D.

But recycling must not only be a nutrition strategy to cope with low nutrient supply through weathering or atmospheric deposition as recycling is also considered as the major nutrition mechanism (e.g. fast turnover of organic matter minimizes nutrient loss from the system). Importantly, there seems also to be a misunderstanding on the organic and geogenic pathway presented in Uhlig and von Blanckenburg 2019. This is, because both pathways operate simultaneously but in terms of fluxes (and on the short timescale) the organic pathway is way more important than the geogenic pathway which becomes essential on the centennial to millennial timescale. Given these complexities in ecosystem nutrition the authors are not putting their findings well in context of ecosystem nutrition.

We recognize that in the first version of our manuscript there was a bit of a "leap in interpretation" regarding the concept of "geogenic" *vs.* "organic" pathways of Uhlig and von Blanckenburg (2019) that was not doing justice to the original ideas developed by these authors. Uhlig and von Blanckenburg (2019) consider that the "slow" geogenic pathway (nutrient supply by weathering) could result in the compensation for the small, but continuous "leak" of organic matter (and associated nutrients).

[Figure]

Plains:
-Higher GPP, TER, MAP, MAT, residence time
-Lower W, D

Nutrient uptake from rock dissolution (geogenic pathway)

Vegetation

Recycling (organic pathway)

Litter fall

Litter reservoir

Loss of nutrients from leaching and erosion of the litter

Andean Belt:
-Lower GPP, TER, MAP, MAT, residence time
-Higher W, D

Nutrient uptake from rock dissolution (geogenic pathway)

Vegetation

Recycling (organic pathway)

Litter fall

Litter reservoir

Loss of nutrients from leaching and erosion of the litter

In this context, what we think our data and analysis show is that the size of this geogenic pathway (compared to the organic pathway) shifts with the weathering regime as indexed by W/D – and, as pointed out by the reviewer in a previous comment, with MAP and MAT too. As explained in the attached figure, we interpret our data as reflecting a more

"subdued" geogenic pathway in the Amazon plains (corresponding to regions drained by the "dilute" and "main" tributaries in our manuscript), because of low erosion rates (and associated slow litter loss), or high MAP and MAT allowing for efficient nutrient recycling from litter. By contrast, in the Andean belt, high erosion rates and lower MAP and MAT decrease the efficiency of recycling and thus increase the loss of nutrients, thus require a higher uptake from rock dissolution (see figure). To solve the concern raised by the review, the discussion in section 4.4 will be changed to remove any apparent opposition between the "geogenic" and "organic" nutrient pathways.

In addition, we also acknowledge the lack of contextualization with references to previous studies in ecosystem nutrition in the first version of our manuscript. To solve this problem, we will improve our discussion and in particular add relevant references in the discussion in section 4.4 on the ecosystem nutrition (*eg.* Chadwick et al., 1999; Elser et al., 2007; Selva et al., 2007; Cleveland et al.,2011; 2013; Augusto et al., 2017).

Thus, the authors somehow over interpret their findings in section 4.4 because of the generalization made and because the field of ecosystem nutrition is widely ignored (e.g. lacking in the introduction). For this reason, I either recommend down tuning of the conclusion or discussing their relationships in the context of ecosystem nutrition (the latter would likely bias the focus of the manuscript).

We agree with the reviewer, and will consequently tune down our conclusions in this part of the manuscript and will amend the text to clearly show that these interpretations should be seen as hypotheses that should be further tested in future studies.

The section 4.4 has been thoroughly re-written and conclusions deriving from our findings have been down tuned. We clarified the meaning of the "geogenic" and "organic" pathways. We also added reference to a significant body of literature on ecosystem nutrition strategies and now avoid over interpretation.

Moreover, that Ba is so strongly cycled through biota is surprising as to my knowledge Ba is neither essential to plants nor considered as a micronutrient or even as a plant beneficial element, following definitions by Marschner (2011). Thus, the strong cycling of Ba through plants in the giant ecosystem of the Amazon may warrant more attention in the manuscript.

Again, we agree with the reviewer that this aspect was not well discussed in the first version of our manuscript. According to the definitions of Marschner (2011) we will remove the term "micro nutrient" for Ba from the manuscript. Although the biological role(s) of Ba are still far from being elucidated – apart from the apparent toxicity of high Ba amounts for plants (*e.g.* Lamb et al., 2013, and reference therein) – previous studies have shown that its chemical similarity with other rock-derived nutrients enables its uptake by plants. Indeed, Bullen and Bailey (2005) have demonstrated the significant biological uptake of Ba and attributed it to its similar ionic radius compared to other alkali-earth elements (Ca, Sr), as well as likely K. A more recent study has shown that Ba uptake scales with that of Ca even if the exact role of Ba in plants has not been identified (Myrvang et al., 2016). Finally, significant uptake and recycling of Ba by the vegetation have been shown to exist by Bullen and Chadwick (2016) using the Ba isotope composition. Therefore, despite the poor constraints on the specific role of Ba in plants, existing knowledge argues in favor of a "nutrient-like behavior" for Ba. This is the term we plan to use in the revised version of the manuscript, and we will slightly extend the justifications in the introduction as to why Ba can be, to some extent, used as a tracer of other rock-derived nutrients. However, we also acknowledge that the behavior of Ba during uptake and recycling by plants is likely to differ to some extent from that of major rockderived nutrients such as Ca, Mg, K or P (these different major nutrients already differing in their behavior during cycling by ecosystems). To reflect the potential decoupling between the cycling of Ba and that of other rock-derived nutrients, we will clearly state in the introduction that using Ba as a tracer of other rock-derived nutrients should be understood as a "working hypothesis" of the manuscript. We will also tune down, where relevant in the discussion, the generalization we make from the quantification of the Ba cycle only, to the other major rock-derived nutrients.

We removed the term "nutrient" for Ba and use "nutrient like-behavior".

Regarding iii) I basically did not understand why the silicate weathering flux should be corrected for the imprint of the biosphere. This is because mineral nutrients, which were cycled through biota are mainly (apart from atmospheric inputs) of geogenic origin, hence released by chemical weathering then cycled through the biosphere and ultimately appear with some delay in the river. I may misunderstood section 4.7, thus please revise this section for more clarity.

Estimates of catchment-scale $CO_2$ consumption by silicate weathering from river dissolved chemistry rely on the assumption that each milli-equivalent of the major cations released by silicate weathering ($Na^+$, $Mg^+$, $Ca^{2+}$, $K^+$) has found its way to the river, and more particularly to the sampling point. Our study shows that after the release from silicate rocks, some rock-derived nutrients can be taken up by the biota to a significant extent and "go missing" in the rivers. In other words, each milli-equivalent of $Mg^+$, $Ca^{2+}$, or $K^+$ that ends up in the biota or dead organic matter on land, and *remains* there (and thus does not "appear with some delay in the river", as suggested by the reviewer) is absent when the "accounts are made" using river data. Importantly, this effect is important only if the nutrients remain in the (living or dead) organic reservoir, meaning that they are not returned to the river through litter remineralization. As discussed in our manuscript, for nutrients to remain in the organic reservoir, two possibilities exist: (1) if particulate organic matter is eroded fast enough such that remineralization is incomplete, (2) a transient accumulation of nutrients in the organic matter reservoir on land is certainly possible over some time scale (see our reply to the first reviewer comment above), that is difficult to determine. In any case, our discussion in section 4.7 suggests that the common assumption that river solute fluxes constitute a reliable proxy for catchment-scale weathering rates is basically wrong. As a result, the fact that elements in rivers are missing allows us to try to quantify and correct for this lost. The reviewer comment indicates that this was not clearly explained, and we thus will clarify section 4.7 during revision.

The section 4.7 has been deeply re written.

Text: Line by line detailed comments:

L.4 delete "measured" as it is clear that the data was measured

We will delete "measured" in the sentence.

This was done.

L.14 please specify through what Ba is cycling exactly (e.g. biota, secondary minerals)

The sentence will be reworded and we will point out that Ba is cycled through biota.

This was done (line 14).

 L.16 see comment L.14

We will add "biological" in this sentence: "... allow us to discuss the role of erosion rates on the biological cycling of rock-derived nutrients".

This was done (line 16).

L.21 To my understanding "minor nutrient" is not a well-defined term for nutrients. Nutrients are either called "essential" elements for plant growth or grouped into the categories "macronutrient", "micronutrient" and "plant beneficial element". Following Marschner (2011) Ba belongs to none of these groups.

According to Marchner (2011), we will delete the "nutrient" term for Ba. Instead, we will switch to qualify Ba as having a "nutrient-like behavior", as explained above.

We removed the term nutrient for Ba (see above).

L.24 for completeness it would be nice to also tell the reader the "yet reported value"

We will add the estimated value of $CO_2$ consumption through silicate weathering from Gaillardet et al. (1997).

We added the values (line 25).

L.32 "making up our water quality" is out of place as the manuscript is not about water quality

We will delete this part of the sentence.

This was done.

L.48 The reference Napieralski et al. 2019 may be useful here, but I do not think it is the most appropriate one for the general information made here.

We will change this reference.

The reference has been deleted.

L.59 The role Ba plays in plants should be briefly introduced. I also do not think that the Viers et al. 2005 reference is the most appropriate one.

As explained above, we will change the term "minor nutrient" to "showing a nutrient-like behavior", in order to avoid confusion. We will also add references demonstrating the uptake of Ba by the vegetation.

The role of Ba in plants has been added (lines 72-79) and the reference has been deleted.

L.88 should read "drainage area"

We will add "area".

This was done (line 105).

L.90 Mean annual temperature (MAT) and mean annual precipitation (MAP) should be provided as they may be important for section 4.4

We will add information about the MAT and MAP of each basin in this section and discuss the role of this parameters in discussion section 4.4 as well (see our answer to the main concern ii) above).

This was done (section 2.1).

L.96 Providing data on erosion rates (e.g. ranges) would help the reader to assess how high or low the rates really are.

L.105 see comment L.96

We will add the range of erosion rates for each geographical unit of the Amazon Basin, based on the numbers reported by Wittmann et al. (2011).

This was done (section 2.1).

L.117 Please remove "are from . . . repository" as it is not important to tell where samples are stored

We will remove this.

This was done.

L.122 It reads that cation concentrations were measured by ion chromatography. Please revise if incorrect.

This is correct.

L.140 Please provide the typical sample weight used for digestion.

The quantity of solid samples requires is only a few mg. Nonetheless, we usually digest much higher quantities in order to perform several other isotope analyses. We will clarify this in the text.

This was done (lines 164-165).

L.145 Which matrix elements were eluted with 2.5 M HCl and how many mL of 2.5 M HCl and 6M HCl were used?

We consider that almost all of the matrix elements (Na, K, Mg, Ca and to a lesser extent Al, Fe and Ti for solid samples) are eluted with this step. We will add this information. We will also provide the number of mL used for each elution step.

This was done (lines 169-170).

L.146 What is meant by "carried out sequentially twice"? Needs rewording

The protocol of separation was applied twice (that is in series, not in parallel) to ensure a total purification of Ba. We will reword this sentence for clarity.

This was done (line 171).

L.152 "(up to 300 ng)" can be removed as this information is included in the range 250-300 ng L.153 "coupled to" should be replace by "using"

We will remove this information and replace "coupled to" by "using".

This was done (line 178).

L.154 I am wondering why so many cups were used. Please explain why each of the measured isotopes were needed for data evaluation. The ones which are measured but not used for data evaluation must not be mentioned.

Collecting all isotope during measurements allows us to ensure that all isotope ratios follow mass-dependent fractionation and in particular that there is no isobaric interference affecting our measurements (see answer below).

L.163 I am wondering why the delta 138Ba was calculated from delta 137Ba and not from the measured isotope ratios 138Ba/134Ba (as is done for delta 137Ba by using measured 137Ba/134Ba).

Small amounts of Ce and La remaining in the Ba eluates (for solid samples) can produce isobaric interferences on $^{138}Ba$, thereby biasing the $^{138}Ba/^{134}Ba$ ratios. Thus we measured the $^{137}Ba/^{134}Ba$ ratios and convert them to $^{138}Ba/^{134}Ba$, assuming mass-dependent fractionation (which was checked using the relationship between $^{137}Ba/^{135}Ba$ vs. $^{137}Ba/^{134}Ba$, as explained in our Fig. S2).

L.175 To allow the reader to evaluate how the data on reference materials measured in this study compare to literature values please also provide the previously published delta values.

We will provide these values.

Values have been provided (line 201).

L.253 To conclude that Ba behaves like a nutrient solely from the correlation of K/Na and Ba/Na is not very reasonable. Provided that stream water integrates over all critical zone processes this correlation could have several reasons. Importantly, K is much more soluble when rainwater passes through the canopy (throughfall) than other mineral nutrients. Thus, I am wondering whether Ba is also as mobile as K from the canopy (apparently not as the authors find that Ba is exported in particulate organic form or accumulates in the Amazon biomass).

We agree with the reviewer that this correlation is no proof of the "nutrient-like" behavior of Ba, but believe it is an interesting additional indication (and a novel one compared to those already pointed out in previous studies, see our reply to main comment ii) above). Indeed, in

the critical zone K is affected by both secondary, soil-forming phase (clays, oxides) formation and biological cycling as well. The fact that river dissolved Ba/Na* and K/Na* ratios display a positive relation simply suggests that Ba is affected by the same processes. However, combined with other observations, a specific role of biological uptake in driving this relationship can be inferred. For example, river dissolved Ba/Na* and K/Na* ratios are lowest in the Andean tributaries, where the role of secondary phase formation is limited (as inferred by Li isotopes; Dellinger et al., 2015), leaving biological uptake a very likely explanation for the low river dissolved abundance of these two elements. On the other end of the spectrum, the high river dissolved Ba/Na* and K/Na* ratios found for rivers draining the shields and lowlands suggest than the leaching of organic matter can be a source of Ba and K - instead of a sink - as these rivers display a higher Ba abundance than expected, and this, albeit a smaller extent of secondary phase formation (see Dellinger et al., 2015). We will make sure during revision that this is made clear.

This part has been re written (lines 281-284).

L.272 I am wondering whether the term "model" is appropriate here as later in the manuscript (e.g. L.342) the authors call it "mass balance approach" which I think is the more appropriate term.

As suggested by the reviewer, we will change the term "model" to "mass balance approach", as model seems the more appropriate terms. However, we still believe that generally speaking the conceptual tool built here falls under the class of "models".

We changed model to mass balance approach whenever it was necessary.

L.330-335 Given that D is E+W this section would benefit from saying when E (or W) is high or low.

Also, we will clarify which one of E or W increases with D (as explained in our replies to the reviewer's main comments above).

Clarification has been added (lines 368-370).

L.340 Please split the sentence into two sentences. The first should be on Ba, the second on Li. Also please say that Li has to date not been shown to fractionate during plant uptake.

We will split the sentence into two different parts and will add the relevant reference mentioning that Li is not taken up by the biota.

The sentence has been split into two parts (line 376).

L.343 I do not really see "correlations" in figure 5. Thus, either revise the wording or show a figure that illustrates correlations.

The relation actually exists, but is difficult to appreciate because data are presented as a bar chart. Nonetheless, we will change the term "correlations" to "positive relations" instead.

We changed the terms "correlations" into "relations" (lines 380-386). Please note that we changed the figure 5 to scatter plots.

L.346 Another explanation could be that Ba is in principle less soluble than other elements (as stated earlier in the manuscript).

The term "soluble" as classically used in geochemistry (unlike in chemistry) has a mostly empirical meaning, simply reflecting the high proportion of a given element that is found in the dissolved load (of *e.g.* a river) compared to the total load. Therefore, a "low Ba solubility" is not incompatible (and actually is rather in line) with "processes" being significant drivers of the Ba cycle. However, we will remove this term from the discussion to avoid confusion.

The term "soluble" has been replaced whenever it was necessary.

L.348 To me fˆBa_bio is not "fairly constant". It simply does not show a trend but the fˆBa_bio data is too different to say that they are constant.

We agree that the more accurate wording should be "does not show any trend". In the next version, we will reword the sentences in that way.

We changed the sentence according to the reviewer comment (line 385).

L.349 see comment L.343

As answered above (please see L.343 comment), the relation is hard to appreciate because of their representation in a bar chart. Again, we will change "correlations" to "trends".

We changed the term "correlations" into "relations" (lines 380-386). Please note that we changed the figure 5 in plots.

L.351 Please use another word for "automatic". I do not understand what the authors want to say.

As the three parameters plotted in fig. 5 sum to 1, any change in one of them affects to some extent the other ones (in other words, the "closure constraint" induces a negative covariance between the pairs of "f-terms"). We will use "spurious" instead.

We clarified the sentence (line 388).

L.355 typo, should read "eq. (10)"

We will correct this mistake.

We corrected the cross reference of the equation (line 392).

L.357 Please remove "that we convert in . . ." as this conversion is trivial.

Following the reviewer's comment, we will remove this part of the sentence.

The part of the sentence has been removed.

L.384 "rock supply rate" means "regolith production"? Please clarify.

In soil-mantled landscapes, "rock supply rate" indeed refers to "regolith production". However, river material reflects the export of dissolved and solid material not only from landscape covered by proper regolith and soil, but also from bare bedrock (*e.g.* landslide scars). This is why we think that "rock supply rate" is a more adequate term in the context of our river-based study.

According to our previous answer, we kept this term.

L.399 Beginning from here I got confused and I think one of the reasons is that the authors should be more precise when presenting low or high W/D ratios. This means the reader needs to know that high or low W/D ratios are due to high or low W or E (D= W+E). This differentiation is quite important in order to understand why both GPP and TER have a positive relation to W/D. As explained in the main comments it is somehow surprising that GPP (nutrient uptake) and TER (nutrient recycling) increase with increasing W/D.

As explained above in our replies to the reviewer's main comments (1) many previous studies have explored the relationships between W, D, and W/D (as already discussed at the beginning of the section 4.3 of the first version of our manuscript), and (2) we think that positive relations between W/D and both GPP and TER are consistent with the spectrum of weathering regimes and ecosystems in the Amazon, ranging from the Andes to the Amazonian rain forest plain. Nonetheless, we will reword this part of the manuscript in order to improve clarity.

L.410 To my understand TER does not say anything about the size of biota, if any, GPP would do so. Please revise.

We will rephrase this sentence for clarity.

The sentence has been revised (line 453).

L.413 As briefly explained in the main comments there seems to be a misunderstanding on the organic and geogenic pathway presented in Uhlig and von Blanckenburg 2019.

We answer to this comment above in our replies to the reviewer's main comments.

The ambiguity on geogenic and organic pathways has been corrected (see section 4.4).

L.422 "made available through denudation" Please specify whether E or W in denudation is meant here, as both can have different effects on plant nutrition.

As explained above, W and D increase in conjunction. It is thus difficult to disentangle the relative role of these two fluxes in "feeding" vegetation. However, we will clarify the sentence through *e.g.* "made available through sustained rock supply and subsequent weathering".

We changed the sentence (line 458).

L.424 Please specify which elements are close to Ba in terms of their nutritional role.

As answered to the main comment ii) regarding the possible role of Ba in plants, we will remove any part that would lead the reader to think that Ba is a "nutrient" and qualify Ba as having with a "nutrient-like" behavior, as suggested by the literature.

L.426ff I doubt that larger pools of bio-available Ba "enhance" Ba uptake. First, the bio-available pool must be continuously replaced over time and second given that Ba is not a nutrient with high bio-available Ba pools more Ba might be taken up but uptake not enhanced. Hence, please revise the wording. I am also wondering why Ba should first be incorporated and then taken up from secondary weathering products. My understanding is that is less energetic to plants to take up dissolved nutrients. As Ba, which was incorporated into secondary products was already dissolved, I would assume plants should have utilised Ba before its incorporation into secondary products.

The reviewer has clearly identified two weaknesses in this potential interpretation. We agree that this scenario constitutes an unsupported interpretation for the observed relationships, and does not add much to the manuscript. Thus we will remove it from the manuscript.

We removed this interpretation.

L.433 Please check whether water residence time in the critical zone is longer at steeper slopes. To me the opposite makes sense.

Although counter-intuitive, a longer water residence time below steeper slopes is the correct interpretation proposed in Torres et al. (2015) based on the isotopes of the water molecule. Indeed, following these authors deep, clay-rich soils in the plains might prevent water infiltration allowing for the rapid transfer of water to the streams during / after precipitation events. By contrast, in mountainous areas the presence of fractured and jointed rocks (because of faulting due to tectonic activity) at shallow depths below ground allows for the formation of large rock-hosted aquifers, which in turn results in longer water residence time.

L.473 Please specify what is meant by an "intensive" metric.

To us this basically means that the metric has no additive properties. We will remove this term for clarity and specify what we exactly mean by that.

The term has been removed.

L.478 To assess whether there is really a difference between wˆBa_fluxes and (Ba/Th)n uncertainties would be helpful which are missing in figure 8a. Also, provided that wˆBa_fluxes relies on Q and D (equation C6) integrating over much different timescales I am wondering whether the differences in figure 8a could also be explained by time scale issues.

This is a fair comment, and we will provide uncertainties for each of the parameters plotted in fig. 8, not only $w^{Ba}_{isotopes}$. We already discuss time scale issues in our replies to the reviewer's main comment i) above.

Uncertainties has been added.

L.534 fˆBa_miss is a relative flux and should be named as such to not confuse the reader as the reader is anyhow busy in differentiating when absolute and when relative fluxes are presented.

We will add line 534 (and elsewhere, where relevant) the term "relative" when we talk about the missing flux, but note that $f^{Ba}_{miss}$ was already named a "relative flux" (line 529).

The term relative has been added whenever it was necessary.

L.536 should read "with the relative amount" as fˆBa_bio is a relative flux.

As said above, we will add the term "relative" where $f^{Ba}_{bio}$ is mentioned.

The term relative has been added whenever it was necessary.

L.543 Please let the reader know the time scale.

A discussion on the typical parameter timescales will be given (as discussed above in the replies to the main comment i)).

L.552ff I am surprised about the groundwater discussion as to my understanding rivers, particularly at baseflow, are mainly fed by groundwater. Or do the authors mean an over regional groundwater aquifer contributing to streamflow?

Yes, this is exactly what we mean. Deep aquifers might exist in the subsurface of the Andes or of the Amazon plain (Hu et al., 2017; Frappart et al., 2019), resulting in a water and solute flow that is not accounted for by collecting water at our river sampling locations. We will clarify this.

We clarified the sentence (line 609).

L.585 Either the C in the left-hand term (FˆC_miss) should be removed or *[C]_org added to the right-hand term in equation 15. Otherwise the equation is wrong.

L.586 see comment in L.585

This is true. We will add a term "[C]_org" term in the equation.

We have corrected the equation (see eq. 15).

L.591 please add "particulate" to "organic carbon"

We will add "particulate" to the sentence.

We changed it (line 643).

L.609ff The section heading reads that it is valid for all mineral nutrients. Please add "Ba" in order to avoid a generalization which may not hold for other mineral nutrients. Figure 11 could be called 11a and 11b which would allow the authors to cross reference each scenario mentioned in section 4.6 directly to the scenarios shown in figure 11. Please see again major comments (ii) for revision.

We will change the title of this section to avoid over-interpretation and over-generalization. We will also add panel letters to fig. 11.

We changed the title.

L.615 "new" Ba is a formulation from Cleveland et al. (2013) (new P), thus please cite.

We will add this citation.

The citation has been added (also several times in the text).

L.628 Please see major comments (iii). Moreover, I do not see a clear trend in figure 12, thus high or low R_sil+bio/sil should be treated with caution.

We agree, and will modify the text to account for the fact that no clear trend in visible in fig. 12. However, we note that this ratio is higher than 1 for many rivers, which is the most important finding in this section.

The figure has been removed and discussion has been tune down.

L.636 "the uptake flux of this element" is misleading because [X]_bio represents a concentration in equation 17.

"[X]_bio" effectively corresponds to a concentration and not to a flux. We will fix this issue in the text.

We corrected this mistake.

L.702 better use "assess" instead of "appraise"

We will change "appraise" to "assess".

We changed it (line 747).

L.707 "atm" in "(Ba/Cl)_atm" should be replaced by "seawater" as according to the text it represents the seawater ratio?

Strictly speaking, the Ba/Cl ratio of atmospheric inputs can differ from that of seawater as dust dissolution (in rain droplets, or in soils after deposition) or biogenic components (Boström et al., 1989) might increase Ba over Cl in this contribution (although this is something we assume to be negligible here, or at least that such dust would be derived from within the considered catchment, such that its contribution could enter the "rock weathering" component, as classically made in river studies). Regarding particles with biogenic origin, their dissolution is taken into account in our isotope mass balance approach, as the latter considers the net difference between uptake and dissolution. Therefore, "atm" is a more generic term than "seawater" to characterize this contribution. As a consequence, we would like to keep using this term in the next version, but will clearly mention that we assume that (Ba/Cl)_atm is equal to the Ba/Cl of seawater.

Figure 1 Axis caption "latitude, longitude" is missing. Also, the left-hand part of the altitude scale can actually be removed as information on the altitude is more relevant for the sample locations.

We will add this caption and remove the left-hand part of the altitude scale.

This was done.

Figure 2 Please add the per mil sign next to delta 138Ba

We will add the per mil sign.

This was done.

Figure 6 I am wondering why "bio" and "sec" show trends for absolute fluxes when plotted against D or W/D in Figure 6 but not for relative fluxes in Figure 5. Provided that in this manuscript so many equations, ratios and ratio over ratios are presented please explain this difference. Regarding style, please frame the legend. Instead of using "+" in this and the other figures please write "increasing" or "decreasing" as the plus is confusing. Also, please remove the arrow and "Biological uptake of rock-derived nutrients" as FˆBa_bio only shows Ba and no other elements and as this information does not provide additional information that the axis label and legend itself.

As explained in one of our reply above, because the three relative proportions sum to 1 they are to some extent automatically anti-correlated. When scaled to dimensional fluxes using a common factor, they are likely to show some degree of correlation, as shown in fig. 6. In our case, the relative proportion $f^{Ba}_{diss}$ decreases while $f^{Ba}_{sec}$ increases with an increase of D. As these two parameters are changing to larger extent than $f^{Ba}_{bio}$, the latter does not really change with D. As a result, to further examine the spatial variability in Ba biological uptake, we need to turn it into a dimensional flux.

We will also change the layout of the figure according to the reviewer's comments.

This was done. Please note that figure 5 has been turned into scatter plot for each $f^{Ba}_i$ term.

Figure 7 In order to better assess whether the trends in this figure are real I would like to see uncertainties. The arrows and text next to the y-axis of panel a and b can be removed as they do not provide additional information. In panel d, I believe there is a misunderstanding regarding the "organic" and "geogenic" nutrient pathway as in eroding systems both are present. The main pathway is the organic one (most important on time scale of up to a few years only) and the geogenic one is minor in terms of fluxes but most important on the centennial to millennial timescale. Thus, it makes no sense to put FˆBa_bio in dependence of both pathways in this figure.

We will provide uncertainties on the Ba-based parameters, and change the layout of the figure according to the reviewer's comments. As answered in the replies to the reviewer's main comments, we will fix the apparent misunderstanding in the concepts around the "geogenic" and "organic" pathways, both in the text and in this figure.

This was done.

Figure 8 In the figure caption it says "Uncertainties on wˆBa_fluxes" but in panel 8a no uncertainties are shown. Is this correct or are the uncertainties smaller than symbol size. This needs clarification. Regarding uncertainties I also noticed that the error bars are differently long towards positive and negative directions. Please check.

We will provide uncertainties on these parameters in the next version. The difference between negative and positive error bars stems from the fact that uncertainty here was calculated by Monte-Carlo simulations, which allows for non-symetric (in particular, non-normal) probability distribution of the calculated parameters (this is because the calculated parameters results from multiplicative operations between variables that are assumed to be normally distributed - multiplication between normally-distributed parameters does not result in normally-distributed parameters).

Uncertainties has been added.

Figure 9 The arrows next to the x-axis and y-axis do not provide additional information and should be removed. Latest with this figure I would like to see a comparison of the $F^{Ba}_{bio}$ with a more direct estimate of the Ba uptake flux, e.g. based on GPP and Ba concentrations in plants.

We will remove the two arrows of the x and y-axis of Figure 9. Nonetheless, we don't think that providing $F^{Ba}_{bio}$ based on GPP will add more information, as discussed in our replies to the reviewer's main comments. However, we plan to add a short piece of text discussing of how this comparison informs on the Ba content of the exported organic matter across the Amazon Basin, and this compares to independent estimates of the Ba content in biological material (as estimated from our own compilation), as explained above.

We removed the arrows according to the reviewer comment.

Figure 11 Please name both panels (e.g. "a" and "b"), which allows the authors to cross reference the figure more detailed in the main text (section 4.6). Also in the figure caption tell the meaning of the arrow sizes.

We will name each panel to avoid confusion. Also, we will change the arrow size accordingly to size of the corresponding fluxes, and explain this in the caption.

This was done.

Answer to the Reviewer #2.

We thank the Reviewer #2 for his/her constructive comments on our manuscript. As raised by the other reviewer, one of the major issue of the first version of our manuscript is the inappropriate use of the "nutrient" term for Ba. We will thus significantly rewrite some part of the manuscript to remove this confusion. Additionally, Reviewer #2 mentions some lack in the graphical report of uncertainties for some parameters we used in the manuscript. We will solve this issue by adding error bars when necessary (see the answers to the specific comments below).

The manuscript is well written and organized and a reader can follow the authors' argumentation. The English of the second version of the manuscript benefited greatly from proof reading after the access reviews. I encourage the authors to also check the supplement for English grammar and style. I recommend publication of the manuscript in *Biogeosciences* after moderate revisions. My comments, which I hope the authors will find useful and constructive, are listed below.

We are pleased that the reviewer found improvement in English grammar and style. During revision, we will further improve the English in the parts mentioned by the reviewer, and everywhere necessary.

Specific comments:

The main conclusion of the study is that a considerable amount of Ba dissolved from the bedrock and transported by the rivers is taken up and stored by biota. I was wondering how exactly plants and/or (micro)organisms utilize Ba. To my knowledge, Ba is not considered an

important nutrient. The authors even state that Ba could be a limiting factor for biota growth (page 19, line 424). This has to be further discussed.

This comment has also been made by the Reviewer #1. We will remove any wrong use of the term nutrient (*i.e.* according to the definitions of Marschner (2011)). The statement made that Ba can be a limiting nutrient is inappropriate, and will be removed in the next version of the manuscript. Therefore, we will remove the term "micro nutrient" for Ba from the manuscript. To answer the reviewer's question, we note that although the biological role(s) of Ba are still far from being elucidated – apart from the apparent toxicity of high Ba amounts for plants (*e.g.* Lamb et al., 2013, and reference therein) – previous studies have shown that its chemical similarity with other rock-derived nutrients enables its uptake by plants. Indeed, Bullen and Bailey (2005) have demonstrated the significant biological uptake of Ba and attributed it to its similar ionic radius compared to other alkali-earth elements (Ca, Sr), as well as likely K. A more recent study has shown that Ba uptake scales with that of Ca even if the exact role of Ba in plants has not been identified (Myrvang et al., 2016). Finally, significant uptake and recycling of Ba by the vegetation have been shown by Bullen and Chadwick (2016) using the Ba isotope composition. Therefore, despite the poor constraints on the specific role of Ba in plants, existing knowledge argues in favor of a "nutrient-like behavior" for Ba. This is the term we plan to use in the revised version of the manuscript, and we will slightly extend the justifications in the introduction as to why Ba can be, to some extent, used as a tracer of other rock-derived nutrients. However, we also acknowledge that the behavior of Ba during uptake and recycling by plants is likely to differ to some extent from that of major rock-derived nutrients such as Ca, Mg, K or P (these different major nutrients already showing various behaviors in ecosystems). To reflect the potential decoupling between the cycling of Ba and that of other rock-derived nutrients, we will clearly state in the introduction that using Ba as a tracer of other rock-derived nutrients should be understood as a "working hypothesis" of the manuscript.

We removed the term of minor nutrient of Ba. Instead, we used the term "nutrient like behavior" as a working hypothesis.

Did the authors propagate uncertainties of single parameters in their models? Most figures do not have error bars and it is thus difficult to assess whether apparent trends are real or not within uncertainty. Furthermore, I am missing estimations on the uncertainties of, for instance, the 20% underestimated $CO_2$ consumption, the main impact this study might have.

For some parameters such as $f^{Ba}_{bio}$ (eq. 8) or $w^{Ba}_{isotopes}$ (eq. 12), uncertainties were already propagated (Tab. S2, Figs. 8b,c) using a Monte Carlo method. However, the reviewer is right when noting that some uncertainties were not graphically displayed in the first version of manuscript, such as $w^{Ba}_{fluxes}$ and (Ba/Th)N (Figs 8). We will display all these uncertainties in the next version of the manuscript.

We propagated uncertainties whenever it was necessary.

For the isotope mass balance in section 4.2 the authors assume congruent dissolution of the bedrock, i.e., no Ba isotope fractionation. Assuming this assumption is wrong, how large would be the impact of isotope fractionation during rock dissolution on the model output? Would it be negligible?

First, we would emphasize that most of the current batch and/or open flow-through model assume that the dissolution of rocks operates in a congruent manner (see Bouchez et al. 2013; Dellinger et al. 2015). This is confirmed by a wealth of experimental work, for example

by Ziegler et al. 2005 (Si), or Wimpenny et al. (2010) (Mg and Li). Nonetheless, a way to take into account the uncertainty associated to potential incongruent dissolution of the bedrock is to consider the variability in Ba isotope composition in the average bulk rock undergoing weathering (this variability being due to different mineralogical compositions and different Ba isotope composition between these minerals). As our analysis already takes into account the variability of bedrock Ba isotope composition through the term $\delta^{138}Ba_{rock}$ of eq. B3 (equal to -0.02 ± 0.04 to 0.07 ± 0.02), we assume that incongruent dissolution – if any – is included within the uncertainties we present for the obtained parameters.

In section 4.4 the authors describe an apparent trend in biological Ba cycling with ecosystem dynamics (Fig. 7c,d). As the figure is now, I fail to see the trend. The only obvious is that the Madeira tributaries have lower GPP and TER values than the rest. However, there is a discrepancy between GPP data in Fig. 7a and 7c. Also, error bars are missing.

We acknowledge that the relations in Fig. 7c,d are not very clear. But given the scope of our manuscript, we expected that the readers would like to see these figures, and this, regardless whether or not GPP and TER show clear relations with $F^{Ba}_{bio}$. Nonetheless we note that, even if the relation is weak, we can clearly see the distinction between rivers group units *ie*. Andean tributaries, Main tributaries and "dilute" tributaries. We will rephrase the discussion around this figure in this way. We will add error bars on the model, as the reviewer suggests.

We removed the term "trend". Instead, we now discuss the broad difference between Andean and Plain tributaries (see section 4.4 and Fig. 7).

In section 4.7, I do not agree with the authors' interpretation that $R_{(sil+bio)/sil}$ increase with very low W/D, based on Fig. 12. The argumentation is apparently based on one data point. Also, this figure lacks error bars.

We will remove this argument based on one data point, and provide error bars.

The uncertainties has been added (see table S.8) and discussion on these data has been greatly mitigated.

Appendix B: The authors estimated the Ba isotope fractionation between dissolved Ba and Ba taken up by biota, admitting that it is poorly constrained. Yet, they state the fractionation with a fairly high precision of ±0.05 ‰. How reliable is the estimated fractionation?

We acknowledge that the uncertainties on this parameter was underestimated in our present analysis. Actually, the number of data on vegetation is very scarce but consistently show negative values (resulting in an estimate of the fractionation factor associated to biological uptake between -0.25 to -0.75). However, although its exact value is under-constrained, the fact that this parameter is negative is the main driver of our findings. To show

this, we plot in the attached figure the computed $f^{Ba}_{bio}$ of the manuscript against $f^{Ba}_{bio}$ calculated using the two extreme values we can estimate from the literature for the isotopic fractionation for biological uptake ($\Delta_{bio\text{-}diss}$ of -0.25 to -0.75). We note that by doing so, each $f^{Ba}_{bio}$ shows positive correlation, thus leaving the trends shown in our manuscript unchanged, and lending confidence in our interpretations. Nonetheless, during revision we will give to the $\Delta_{bio\text{-}diss}$ a more realistic uncertainty, and modify the resulting error bars on $f^{Ba}_{bio}$ (and all derived parameters) in the relevant figures accordingly.

The uncertainty on $f^{Ba}_{bio}$ (obtained through Monte Carlo simulations) has been re-calculated assuming a uniform distribution for $\Delta_{bio\text{-}diss}$ between the two extreme values ($\Delta_{bio\text{-}diss}$ of -0.25 to -0.75). We note that it does not change our findings and conclusions.

The authors made a great effort in computing and quantifying data and parameters. However, not all derivations of equations can be followed easily. For instance, I failed to understand how equations C5, D5 and D6 are derived given the provided information.

We will provide further details for the derivation of these equations.

We added details in the derivation of the equations.

Page 3, line 59: $^{130}$Ba is a primordial nuclide and can be considered stable under geochemical aspects.

We will mention the fact that the very long decay of $^{130}$Ba allows to consider it as a "stable" isotope.

We added it (line 69).

Page 5, line 134: What are plutonic rocks in this case? What is their lithology?

We consider that these plutonic rocks are mostly granites (felsic igneous rocks; see Stallard and Edmond 1981).

We added granites (line 167).

Page 10, line 214: Please define * in the main text, not only in the figure caption and supplement.

We will add the definition of "*" in the main text.

* has been defined in the text (line 270).

Page 19, line 433: Why is the residence time of water longer along steeper slopes?

Although counter-intuitive, a longer water residence time below steeper slopes is the correct interpretation proposed Torres et al. (2015) based on the isotopes of the water molecule. Indeed, following these authors deep in soils in the plains might prevent water infiltration allowing for the rapid transfer of water to the streams during / after precipitation events. By contrast, in mountainous areas the presence of fractured and jointed (because of faulting due to tectonic activity) and presence of fractures at shallow depths below ground allows for the formation of large rock-hosted aquifers, which in turn results in longer water residence time.

Page 29, line 654: I could not find any data/figure supporting the argument that mainly K weathering flues are influenced by biological cycling. If they are to be found, e.g., in the supplement, please refer to it. Otherwise data have to be provided.

We acknowledge that the influence of biological uptake on the K cycling was somehow too implicit in the text. Indeed, the reason leading us to think that K deriving from rock weathering is strongly affected by biological cycling (Chaudhuri et al., 2007). In addition, when significant uptake by vegetation is found (as inferred by our Ba isotope mass balance), it appears that the addition of K "stored" in the vegetation represent in average around 40% of the K release from rock dissolution. We will clarify the text in that way.

Please see Table S.8.

Page 30, line 669: Please quantify this significant uncertainty!

We will add uncertainty on these data (and the figure as well).

We quantified uncertainties using Monte Carlo simulation.

Page 31, line 672: [...] to be source mainly from silicate rocks [...]

We will add "mainly" to the sentence.

We added mainly (line 715).

Page 33, line 754: Charbonnier et al. (2018) is a review paper. When literature data are used, please cite the original publications (also later in that appendix).

We will add reference of the original publications and where it is necessary.

We added the initial references whenever it was necessary.
Technical comments:

Fig. 7: GPP data are different in panel a) and c)!

We fixed the mistake (see the comment on the lack of relation on this figure above).

We changed the figure.

Fig. S2: Are the error bars correct? They show approximately ±0.15 ‰ on $\delta^{138/134}$Ba. Long-term precision for BaBe27 and JB-2 is however given as ±0.08 ‰.

These error bar represents the uncertainties for each single measurement and not the S.D value or the confidence interval (CI95%).

---

## Author Response (AR2)

Dear Dr Charbonnier and colleagues,

After reading your manuscript and the suggestions by Reviewer #1, I have decided to accept your publication, subject to minor revisions. Please have a look at the reviewer's suggestions for necessary corrections. Your manuscript is very long with 11 different figures. Please consider whether it is possible to move some of your figures, and maybe also parts of your discussion to the appendix.

Best regards,

Edzo Veldkamp

We thank the editor for giving us the opportunity to publish our manuscript in Biogeosciences. Regarding the size of the manuscript, all of the technical discussion had already been moved to the Appendices and Supplementary Material during the previous round of revision, hence it seems hard to find other parts of the text that are easily removable. Therefore, we would prefer to keep the text as it is, especially given the fact that this text is the fruit of an extensive discussion with the reviewers. Nonetheless, during this last round of revision we moved one figure (Fig. 3 now Fig. S3) to the on-line supplementary material, removed another figure (Fig. 11) that was not necessary for the understanding of the manuscript, and optimized the layout of a third figure (Fig. 4), all in order to shorten our manuscript.

I thoroughly reviewed an earlier version of this manuscript. Given that the authors convincingly addressed all main concerns and almost all minor comments raised by both reviewers, reviewing this manuscript again was a real pleasure. Also, the English is very good now. In cases in which the authors did not follow the reviewer suggestions they provide convincing arguments for their opposite view. All in all, the revisions helped to improve the previous and anyhow very good manuscript. I suggest publication of this manuscript after addressing the only few and very minor comments below.

Cheers, David Uhlig

We thank the reviewer for this constructive comment and we appreciate his open-mindedness on the several opposite views we had during the revision process. In overall, he allowed us to strongly improve the manuscript and especially regarding the nutrition strategy of the vegetation. Below we detail the different technical changes that we made.

L.1 better should read "…element cycles on the Earth surface …"
We changed "cycling" to "cycles" (line 1).

L.4 please add the abbreviation (Ba) next to word barium
We added "(Ba)" (line 5).

L.25 Typo: the multiply symbol reads like a comma and the citation style in the abstract should be checked for journal style
We replaced "." by the multiply symbol and changed the citation style (line 25).

L.49 Typo: long-term
"-" has been added (line 49).

L.70 Please add the half-life time
We reworded the sentence "given its half-life of $10^{21}$ yrs" (lines 69-70).

L.114 Typo: remove spacing before the comma
We removed the space before the comma (line 114).

L.121 "consist of" instead of "consist in"?
We changed "in" by "of" (line 121).

L.145 plural for "concentration" to be consistent with e.g. L. 149
We changed for "concentrations" (lines 145 and 153-154).

L.176 "total procedural blank" instead of "blank procedure" as the latter reads like a method
We changed to "total procedural blank" (line 176).

L.362 Typo: "mm/yr" to be consistent with e.g. t/km2/yr
We corrected to mm/yr (line 362).

L.398 replace "strong" by "high"?
We replaced "strong" by "high" (line 398).

L. 419 Provided that F^Ba_sec is also increasing with D it reads misleading to emphasize the biological uptake flux is this sentence only. This sentence needs a slight rewording
The fact that the absolute flux of secondary phase formation increases at higher denudation rates -excluding very high denudation rates- was already emphasized in Dellinger et al. (2015). Then mentioning that $F^{Ba}_{sec}$ increases as well with denudation rates is the most "striking features" might be seen as an overstatement. Nevertheless, we changed the sentence to "The most interesting and novel observation" (line 419).

L.624 I do not understand the wording "after important deforestation" and suggest the deletion of the word "important".
We removed the word "important" (line 624).

L.649 To my understanding it should read NPP (net primary productivity) in equation 17 because according to Chapin et al. (2012) (page 161, eq. 6.2) NPP = GPP – R_plant where R is respiration. To my understanding this is exactly what the authors mean by NEE = GPP-TER in the manuscript. Thus, there is no need to introduce the "NEE".
We changed the term NEE to NPP here and elsewhere in the text when it was necessary (lines 649-650).

L.651 Typo: "values … differ" not differs
We corrected this (line 651).

L.671 Typo: should read "behavior"
We removed one of the two "behavior" (line 670).

L.674 To me it reads that the recycling strategy causes less uptake of "new" nutrients. But actually, the lack of substantial litter export in the described scenario does not require "new" nutrient uptake by plants. Thus, in my opinion the wording "minimizes" (which I understand as causing a strategy) is not correctly used here and needs rewording.
We reworded the sentence (lines 673-674).

L.702 Please add in brackets "R(sil+bio)/sil" behind "weathering fluxes"
We added "(R(sil+bio)/sil)" after weathering fluxes. (line 701).

L.718 Typo: should read "nutrient cycling" and a point is missing at the end of the sentence.
We corrected "nutrient" and added a point (line 718).

L.761 Provided that isotope fractionation factors (in this manuscript the capital delta) are used I found it quite confusing to read the alpha as a symbol for proportions in appendix A and B and would encourage the authors to use an alternative symbol, e.g. "f" for a relative fraction.
The alpha term as the relative contribution from the different rock dissolution to the dissolved species directly derives directly from Dellinger et al. (2015) paper and tables within it. Therefore, we preferred to keep the same term for the sake of the consistency and clarity for peoples interested in reading this paper and the Dellinger et al. (2015). Moreover, the use of "$f$" as a relative source of dissolved species in the river might be confounded with "$f$" as the extent of processes after rock dissolution (*e.g.* $f^{Ba}_{sil}$ vs $f^{Ba}_{sec}$).

L.775 Typo: should read "… influence of …"
We corrected this (line 776).

L.789 Typo should read "as shown in …"
We added "in" (line 787). Also, we added the values for each (Ba/Na)_0 (lines 787-790).

L.840 Typo: should read "… in detail …"
We corrected this (line 841).

L.871 Typo: please add "eq" to (3)
We added "eq" before (3) (lines 870 and 873).

L.879 Typo: should read "… which is a …"
We added "is" (line 881).

L.881 Typo: should read "dimensional value …". Also, this sentence is difficult to read. I suggest ending the sentence after f^Ba_bio and begin a new sentence.
We changed to "value" line (883) and the sentence has been reworded after $f^{Ba}_{bio}$ "Below, we show how to calculate the net biological uptake flux for major rock-derived nutrients based on $f^{Ba}_{bio}$ computations." (lines 883-884).

Supplement:

L.6 should read "… of the river dissolved load"?
We added "load" (line 6).

L.32 Typo: should read "… stacked to produce one …"
We removed "one" (line 32).

L.35 should better read "… to reconstruct the …"
We changed to "reconstruct" (line 35).

L.40 Typo: should read "… to Li"
We removed "the" before "Li" (line 40).

L.46 preposition missing, should better read "river mass budget equations for the …"
We added "for" before "the" (line 46).

L.52 better should read "… but applied it to Li"
We added "to" before "Li" (line 52).

Reference used:
Dellinger, M., Gaillardet, J., Bouchez, J., Calmels, D., Louvat, P., Dosseto, A., ... & Maurice, L. (2015). Riverine Li isotope fractionation in the Amazon River basin controlled by the weathering regimes. *Geochimica et Cosmochimica Acta*, *164*, 71-93.